# Luminance-Aware Statistical Quantization: Unsupervised Hierarchical Learning for Illumination Enhancement

**Derong Kong**[1]  **Zhixiong Yang**[1]  **Shengxi Li**[2]  **Shuaifeng Zhi**[1]
**Li Liu**[1]  **Zhen Liu**[1]  **Jingyuan Xia**[1†]
[1]College of Electronic Science and Technology, National University of Defense Technology
[2]College of Electronic and Information Engineering, Beihang University

## Abstract

Low-light image enhancement (LLIE) faces persistent challenges in balancing reconstruction fidelity with cross-scenario generalization. While existing methods predominantly focus on deterministic pixel-level mappings between paired low/normal-light images, they often neglect the continuous physical process of luminance transitions in real-world environments, leading to performance drop when normal-light references are unavailable. Inspired by empirical analysis of natural luminance dynamics revealing power-law distributed intensity transitions, this paper introduces Luminance-Aware Statistical Quantification (LASQ), a novel framework that reformulates LLIE as a statistical sampling process over hierarchical luminance distributions. Our LASQ re-conceptualizes luminance transition as a power-law distribution in intensity coordinate space that can be approximated by stratified power functions, therefore, replacing deterministic mappings with probabilistic sampling over continuous luminance layers. A diffusion forward process is designed to autonomously discover optimal transition paths between luminance layers, achieving unsupervised distribution emulation without normal-light references. In this way, it considerably improves the performance in practical situations, enabling more adaptable and versatile light restoration. This framework is also readily applicable to cases with normal-light references, where it achieves superior performance on domain-specific datasets alongside better generalization-ability across non-reference datasets. The code is available at: `https://github.com/XYLGroup/LASQ`.

## 1 Introduction

In low-light environments, images frequently experience degradations like reduced visibility and heightened noise, which hinder subsequent vision-related tasks (1; 2; 3; 4). Low-light image enhancement (LLIE) aims to reconstruct perceptually natural scenes by establishing mappings between low-light and normal-light distributions. However, this problem is fundamentally ill-posed, as natural luminance transitions follow continuous physical processes governed by scene radiance and sensor responses, rather than discrete pixel-level correspondences.

While recent deep learning methods—whether supervised (5; 6; 7; 8; 9) or unsupervised (10; 11; 12)—attempt to model light variations through paired or unpaired training, they inherently overfit to static relationships between low/normal-light domains. Supervised methods rely on pixel-level correspondences in paired data, forcing models to prioritize localized correlations over the physics of gradual luminance evolution. Unpaired approaches, though avoiding direct pairing, still depend

---

Derong Kong and Zhixiong Yang contributed equally to this work ([†] Corresponding author: Jingyuan Xia).

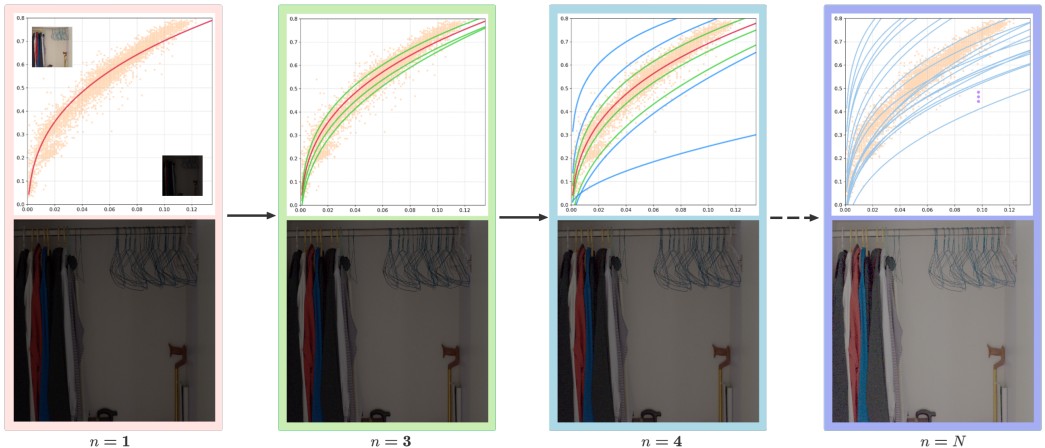

Figure 1: The physics-driven regularity of luminance intensity evolution.

on pseudo-references derived from empirical gamma corrections (10), inheriting prior biases. Both paradigms oversimplify the inherently context-dependent and continuous nature of luminance dynamics, resulting in constrained generalization: models excel in domain-specific scenarios with reference normal-light samples but struggle to adapt to unseen environments or sensor-specific degradations (13; 14). This underscores the necessity for a paradigm shift toward learning luminance transitions from the intrinsic continuity of real-world illumination processes.

This work is motivated by an empirical revelation: natural luminance transitions between low-light and normal-light conditions inherently adhere to power-law density distributions across hierarchical intensity coordinates, as visualized in Fig. 1. Unlike the black-box mappings learned by existing methods, these distributions reveal a physics-driven regularity—pixel intensities evolve along stratified luminance layers governed by cumulative power functions (Fig. 1). Specifically, each layer corresponds to a distinct power-law parameter that dictates how localized or global the luminance adaptation should be. For instance, a single power function approximates uniform luminance adjustment across the entire image (e.g., global gamma correction), while multiple overlapping functions capture spatially varying transitions, mimicking the interplay of scene radiance and sensor responses. By parameterizing these layers through variable-density sampling, where the number of power functions determines the granularity of luminance adaptation, we bridge the gap between pixel-level fidelity and cross-scenario generalization. Denser sampling (more functions) prioritizes localized intensity corrections resembling pixel-wise mappings, whereas sparser sampling (fewer functions) enforces smoother, physically consistent transitions across regions. This hierarchical decomposition fundamentally redefines LLIE: instead of deterministic low-to-normal mappings, we model LLIE as a statistically driven process that progressively traverses luminance layers, emulating the continuous and context-aware nature of real-world illumination dynamics.

In this instance, we propose a luminance-aware statistical framework named Luminance-Aware Statistical Quantization (LASQ) that translates the hierarchical power-law distributions of natural illumination into an adaptive statistical sampling process. We first formulate a scale-adaptive luminance intensity estimation function governed by power-law exponents, where both the base and exponent are dynamically computed from localized intensity statistics across adjustable regions of the luminance map. This function is derived from power-law regularity observed in natural scenes, enabling seamless computation for pixel-level corrections to global adjustments luminance adaptation operators. On the basis of this, we formulate a distribution space for luminance adaptation operators that spans granularity levels—ranging from coarse, scene-wide adjustments to fine-grained, region-specific refinements. A Markov Chain Monte Carlo (MCMC) sampling strategy is then designed to progressively explore this space, initiating from global equilibrium states and iteratively introducing spatially varying layers to simulate the continuum of real-world luminance transitions. The sampled operators follow a Gaussian-like distribution, where high-probability candidates correspond to

physically plausible global adaptations, while low-probability ones represent localized refinements. This naturally reflects the rarity of extreme, pixel-level corrections in natural illumination transitions.

We note that this sampling mechanism is embedded into the forward process of a diffusion model, which learns to traverse luminance layers in an unsupervised manner. By aligning the diffusion trajectory with the hierarchical granularity of luminance adjustments, our framework emulates the gradual, across-scene robust propagation of light in real environments, achieving fidelity-generalization equilibrium through statistically grounded layer-wise enhancement. The proposed LASQ framework thereby attains an optimal balance between local reconstruction precision and global robustness across diverse scenarios, eliminating the need for normal-light reference acquisition. Extensive experiments validate that LASQ, when integrated with a vanilla diffusion model, achieves state-of-the-art performance on non-reference datasets while attaining comparable performance to reference-dependent methods on normal-light benchmark datasets. Furthermore, LASQ exhibits versatile compatibility: it seamlessly adapts to scenarios where normal-light references are available, delivering superior domain-specific enhancement alongside unparalleled cross-dataset generalization capabilities.

Our main contributions are summarized as follows:

- We propose LASQ that fundamentally redefines LLIE by establishing the first physics-aware statistical model grounded in hierarchical power-law luminance transitions. This innovation bridges the gap between physical regularity modeling and data-driven learning paradigms, shifting the LLIE paradigm from deterministic pixel-wise mappings to stochastic processes governed by natural illumination statistics.

- We establish a statistical sampling on hierarchical luminance adaptation operator to emulate illumination transitions, where multi-scale power-law distributions are systematically parameterized and sampled via adaptive MCMC strategies. This enables automatic adaptation from global equilibrium adjustments to localized refinements based on scene-based brightness characteristics.

- We introduce a diffusion-driven learning architecture that systematically incorporates physical illumination priors through progressive luminance layer traversal during the forward diffusion process. This design enables unsupervised hierarchical enhancement while achieving dual-mode compatibility with both reference-based and reference-free scenarios, thereby eliminating dependency on paired reference data.

- Comprehensive experiments show that LASQ achieves i) superior performance on non-reference datasets without any reference guidance, ii) attains comparable performance to reference-based methods on reference-available benchmarks even when references are withheld, and iii) outperforms existing reference-based approaches when references are utilized.

## 2 Related Work

### 2.1 LLIE via Pixel-Level Consistency

Contemporary approaches focusing on pixel-level consistency can be categorized into three evolutionary stages (5; 15; 16; 17; 18; 19; 20; 21). Early works like LLNet (22) and Retinex-Net (23) leveraged paired datasets to train CNNs with pixel-wise losses, achieving precise local corrections but suffering from domain overfitting. Methods like EnlightenGAN (24) introduced cycle-consistency constraints, while Zero-DCE (25) used non-paired training with empirical illumination curves, both inheriting biases from heuristic priors (2). Recent diffusion models (10) enhanced flexibility through noise-to-clean transitions, and FeatEnHancer (1) proposed hierarchical feature fusion to bridge pixel-level and semantic gaps. These methods improved generalization but remained black-box mappings (3). While progressively reducing dependency on strict pixel correspondences, these methods universally prioritize pixel-wise fidelity over the continuous, context-dependent nature of luminance transitions, resulting in constrained generalization when facing unseen scenarios or sensor-specific degradations (13).

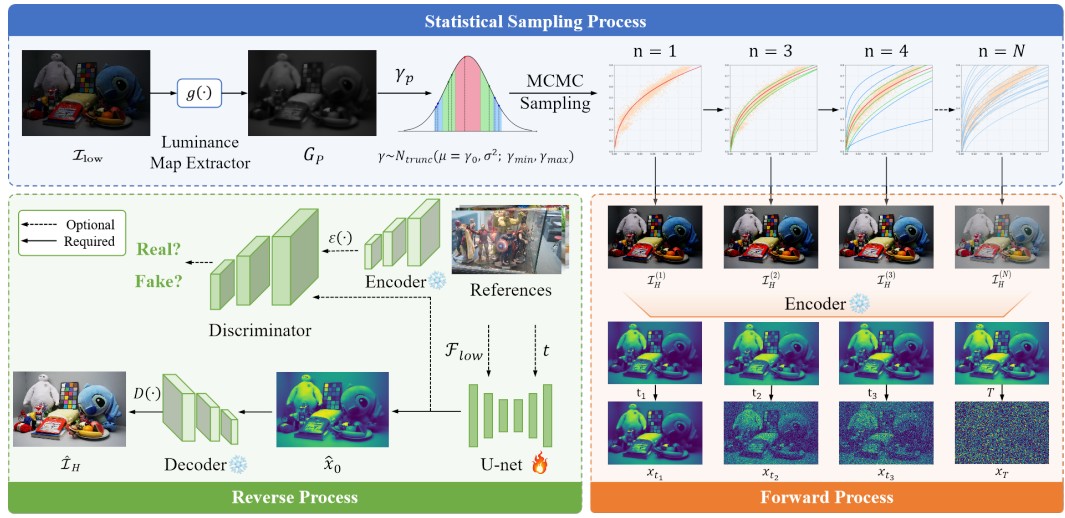

Figure 2: The framework of our LASQ.

## 2.2 LLIE with Illumination Priors

To alleviate these issues, several studies (26; 27; 25) have incorporated illumination-aware correction into deep neural networks, framing LLIE as a curve-estimation problem through different gamma correction samples. Specifically, (27) introduced learnable gamma transforms for illumination adjustment, yet enforced uniform corrections across all pixels. Methods like KinD (26) decomposed images into illumination-reflectance components, but relied on manually designed reflectance priors that oversimplified real-world radiance interactions, and its reliance on downstream pre-training objectives (e.g., perceptual losses) further introduces external biases that impair adaptability to diverse degradation patterns. More latest works like LightenDiffusion (10) integrated Retinex theory into diffusion steps, while GPP-LLIE (28) embedded illumination gradients as diffusion guidance. Though enhancing physical plausibility, these methods still imposed global correction strategies through rigid equation constraints.

Current physics-aware approaches generally ignore the hierarchical power-law distributions that dictate natural luminance changes. These methods, which depend on global gamma-like adjustments or empirical reflectance models, do not adequately represent the layered luminance levels where both localized and global adjustments interact, thereby restricting adaptability to sensor-specific issues and real-world lighting variations.

## 3 Methodology

### 3.1 Notation

The overall framework of our LASQ is illustrated in Fig. 2. At its core, we denote a low-light image by $\mathcal{I}_L \in [0,1]^{H \times W}$ and its normal-light counterpart by $\mathcal{I}_N \in [0,1]^{H \times W}$, and for each pixel index $i \in \{1, \ldots, I\}$ with $I = H \times W$, we write the luminance pair $s_i = \left(\mathcal{I}_L^{(i)}, \mathcal{I}_N^{(i)}\right)$. Besides, $\mathcal{P}$ indicates the image region. We denote the luminance map by $G_{\mathcal{P}}$ and define $\gamma_{\mathcal{P}}$ as our hierarchical luminance adaptation operator. $\mathcal{H}$ and $\mathcal{F}$ represent the hierarchical enhanced image set and latent representations, respectively. Let $\mathcal{E}$ and $\mathcal{D}$ denote the encoder and the decoder. $G_\theta$ and $\mathcal{D}_\phi$ express the generator and the discriminator.

### 3.2 Hierarchical Luminance Modeling

**Luminance Variation Coordinate System:** In preparation for our statistical modeling of low-light image enhancement, we present a unified two-dimensional coordinate system designed to map the

relationship between "normal-light" and "low-light" luminance intensities. Let $\mathcal{I}_L \in [0,1]^{H \times W}$ denote an observed low-light image and $\mathcal{I}_N \in [0,1]^{H \times W}$ be its normal-light counterpart. For each element located at $(x_i, y_i)$, we write $\mathcal{I}_L^{(i)} = \mathcal{I}_L(x_i, y_i)$ and $\mathcal{I}_N^{(i)} = \mathcal{I}_N(x_i, y_i)$, $i = 1, \ldots, I$, where $I = H \times W$. By treating each pair $\left( \mathcal{I}_L^{(i)}, \mathcal{I}_N^{(i)} \right)$ as a point $s_i$ in the plane, we form the Luminance Variation (LV) coordinate system:

$$\left\{ s_i = \left( \mathcal{I}_L^{(i)}, \mathcal{I}_N^{(i)} \right) \right\}_{i=1}^{I}. \tag{1}$$

Empirical observation reveals that, within the normalized range $[0,1]$, the low-light intensities exhibit a heavy-tailed power-law distribution, which can be approximated by a set of power-law functions

$$ax^{\kappa}, x \in [0,1], \kappa > 0, a > 0, \tag{2}$$

where $\kappa$ denotes to the illumination change level by this power-law curve. As depicted in the fourth column of Fig. 1, the asymmetric distribution of underexposed elements can be approximated by a set of sampled power-law curves, providing a rigorous foundation for adjusting the low-light distribution to a specified target using mapping or sampling methods.

**Statistical Sampling Process:** Utilizing the LV coordinate system, we initially introduce the regional luminance scalar $G_{\mathcal{P}}$ for any designated image area $\mathcal{P} \subseteq [1, H] \times [1, W]$. This scalar encapsulates the characteristic distribution of luminance within the region, and its precise formulation is provided in the Appendix. Utilizing Eq. (2), we proceed to compute our hierarchical luminance adaptation operator (LAO) $\gamma_{\mathcal{P}}$ as follows:

$$\gamma_{\mathcal{P}} = (\alpha + G_{\mathcal{P}})^{\beta_{\mathcal{P}}}, \quad \beta_{\mathcal{P}} = 2G_{\mathcal{P}} - 1 + \eta \frac{\sigma_{G_{\mathcal{P}}}^2}{\sigma_{G_{\mathcal{P}}}^2 + \delta}, \tag{3}$$

where $\alpha \in (0, 1]$, $\eta$, $\delta$ are hyper-parameters that control adjustment strength and contrast gain. Empirical analysis reveals distinct luminance adaptation patterns: single LAO exhibits uniform global luminance modulation, while multi-LAO configurations enable region-specific refinement. This phenomenon emerges because curves in the central regime of the power-law distribution (highlighted in red in Fig. 2) demonstrate universal traversal across all LAO set cardinalities, whereas boundary regions (depicted in blue) are exclusively accessible to high-density LAO sets implementing fine-grained adjustments. We consequently model this operator distribution through a symmetrically truncated Gaussian distribution (Fig. 2, top center) as follows:

$$p(\gamma) \propto \exp\left( -\frac{1}{2} (\gamma - \gamma_0)^2 / \sigma^2 \right), \quad \gamma \in [\gamma_{\min}, \gamma_{\max}], \tag{4}$$

where $\gamma_{\min} = \min_{i,j} \gamma_{(i,j)}$, $\gamma_{\max} = \max_{i,j} \gamma_{(i,j)}$ and $\gamma_0 = \frac{1}{HW} \sum_{i=1}^{H} \sum_{j=1}^{W} \gamma_{(i,j)}$. The bounded distribution can be formally expressed as $\gamma \sim \mathcal{N}_{\text{trunc}}\left( \mu = \gamma_0, \sigma^2; \gamma_{\min}, \gamma_{\max} \right)$.

Building upon the symmetrically truncated Gaussian distribution $p(\gamma)$, we devise a hierarchical Markov Chain Monte Carlo (MCMC) sampling scheme to generate LAO sets $\Gamma = \{\Gamma_n\}_{n=1}^{N}$, where each iteration $n$ produces $2^{n-1}$ distinct LAO configurations via adaptive chain transitions, referring to $\Gamma_n = \left\{ \gamma_{\mathcal{P},z}^{(n)} \right\}_{z=1}^{2^{n-1}}$. The MCMC process at the $n$-th iteration is given by:

$$p(\mathcal{I}_H^{(n)}) = \int p(\mathcal{I}_H^{(n)} | \Gamma_n) p(\Gamma_n) d\Gamma_n \approx \sum_{z=1}^{2^{n-1}} p(\mathcal{I}_H^{(n)} | \gamma_{\mathcal{P},z}^{(n)}) p(\gamma_{\mathcal{P},z}^{(n)}), \tag{5}$$

derived from the continuous formulation through discrete sampling approximation. Each trial constructs a Markov chain defined by the transition kernel:

$$q\left( \gamma_{\mathcal{P},z}^{(n)} \mid \gamma_{\mathcal{P},z-1}^{(n)} \right) = \mathcal{N}_{\text{trunc}}\left( \gamma_{\mathcal{P},z}^{(n)} \mid \gamma_{\mathcal{P},z-1}^{(n)}, \lambda^2; \gamma_{\min}, \gamma_{\max} \right), \tag{6}$$

where the step size $\lambda$ adaptively balances exploration-exploitation trade-offs across hierarchy levels.

The dynamically partitioned grid strategy ensures progressive refinement: at iteration $n$, the image is divided into $m_n \times w_n$ non-overlapping patches ($m_n = 2^{\lceil (n-1)/2 \rceil}$, $w_n = 2^{\lfloor (n-1)/2 \rfloor}$). This induces hierarchical luminance-corrected images $\mathcal{H} = \{\mathcal{I}_H^{(n)}\}_{n=1}^{N}$, where each $\mathcal{I}_H^{(n)}$ encapsulates $2^{n-1}$ locally optimized gamma correction patterns. Crucially, every MCMC trial synthesizes a self-consistent LAO set that traverses luminance hierarchies through state-dependent transitions, enabling coarse-to-fine representation learning where global brightness constraints guide local refinements and vice versa.

### 3.3 Hierarchically-Guided Diffusion

**Forward Process with Hierarchical Guidance:** The sampled set $\mathcal{H} = \{\mathcal{I}_H^{(n)}\}_{n=1}^N$ employs stochastic learning via diffusion transitions, exploiting the Markov property—each $\mathcal{I}_H^{(n)}$ relies only on its forerunner—to adaptively direct noise injection. The low-light image $\mathcal{I}_L$ and $\mathcal{H}$ are encoded together using $\mathcal{E}(\cdot)$, incorporating $k$ residual blocks and max-pooling for latent features $\mathcal{F}_L \in \mathbb{R}^{\frac{H}{2^k} \times \frac{W}{2^k} \times C}$ and $\left\{\mathcal{F}_H^{(n)}\right\}_{n=1}^N \in \mathbb{R}^{N \times \frac{H}{2^k} \times \frac{W}{2^k} \times C}$. We align the $T$-step diffusion with $\left\{\mathcal{F}_H^{(n)}\right\}_{n=1}^N$ using a temporal mapping $\psi : \{1, \ldots, T\} \to \{1, \ldots, N\}$, $N \leq T$, by $\psi(t) = \lfloor t \cdot N/T \rfloor$, such that:

$$T = \sum_{n=1}^N |T_n|, \quad t \in T_n \Rightarrow x_0 = \mathcal{F}_H^{(\psi(t))}. \tag{7}$$

The forward diffusion adds Gaussian noise progressively:

$$q(x_t|x_{t-1}) = \mathcal{N}\big(x_t; \sqrt{1-\beta_t}x_{t-1}, \beta_t I\big), \tag{8}$$

where $t \in \{1, \ldots, T\}$ denotes the diffusion timestep, $x_t$ represents the random variable at timestep $t$, and $\beta_t$ denotes the noise variance. Accordingly, for each temporal interval $T_n$, the corresponding spatial variant $\mathcal{F}_H^{(\psi(t))}$ is utilized as the illumination normalization reference, thereby maintaining luminance-consistent forward sampling. By incrementally incorporating spatial luminance awareness from coarse to fine scales into the diffusion forward trajectory, the model acquires a multi-level representation of illumination dynamics. This hierarchical perception facilitates adaptive noise scheduling and enhances robustness across a wide range of lighting conditions.

**Hierarchically-Guided Diffusion Denoising:** During the reverse training phase, the denoising network $\epsilon_\theta(x_t, t, \mathcal{F}_L)$ is trained to achieve:

$$x_{t-1} = \frac{1}{\sqrt{1-\beta_t}}\left(x_t - \beta_t\epsilon_\theta(x_t, t, \mathcal{F}_L)\right) + \sigma_t b, \quad b \sim \mathcal{N}(0, I), \tag{9}$$

where $\sigma$ denotes the standard deviation. Based on the variational lower bound of the forward process, we minimize the mean squared error between the true noise and the network prediction, leading to the simplified noise prediction objective:

$$\mathcal{L}_d = \sum_{t=1}^T \mathbb{E}_{q(x_0, \mathcal{F}_L)}\left[\|\epsilon - \epsilon_\theta(x_t, t, \mathcal{F}_L)\|^2\right], \tag{10}$$

where $\epsilon$ represents the actual injected noise. To ensure overall image smoothness and preserve fine details while minimizing generation artifacts, we employ the global label $\mathcal{F}_H^{(\psi(0))}$ to weakly guide the reverse diffusion process:

$$\mathcal{L}_g = \left\|\mathcal{D}(\hat{x}_0) - \mathcal{D}(\mathcal{F}_H^{(\psi(0))})\right\|_1, \tag{11}$$

where the $\mathcal{D}(\cdot)$ denotes the decoder. During inference, under the guidance of the low-light input $\mathcal{F}_L$, we employ the diffusion model's implicit sampling strategy (29) for reverse denoising. The model utilizes its learned distribution to fit the optimal illumination-enhanced feature representation $\hat{\mathcal{F}}_N$, which is then decoded to yield the final output $\hat{\mathcal{I}}_N = \mathcal{D}(\hat{\mathcal{F}}_N)$.

The LASQ framework can integrate effortlessly with normal-light references. In this setup, an optional adversarial discriminator $\mathcal{D}_\phi$ complements the LASQ, creating a hybrid diffusion-GAN model. The generator's training involves a combined loss:

$$\mathcal{L}_{\text{total}} = \lambda_d \mathcal{L}_d + \lambda_g \mathcal{L}_g + \lambda_{\text{GAN}} \underbrace{\mathbb{E}_{\mathcal{I}_L}\left[-\log \mathcal{D}_\phi(G_\theta(\mathcal{I}_L))\right]}_{\text{adversarial penalty}}, \tag{12}$$

where adversarial training refines high-level textures while preserving the physical grounding from diffusion priors. These results are presented with LSAQ++ in the simulation. Implementation details for reference-augmented training are provided in the Appendix.

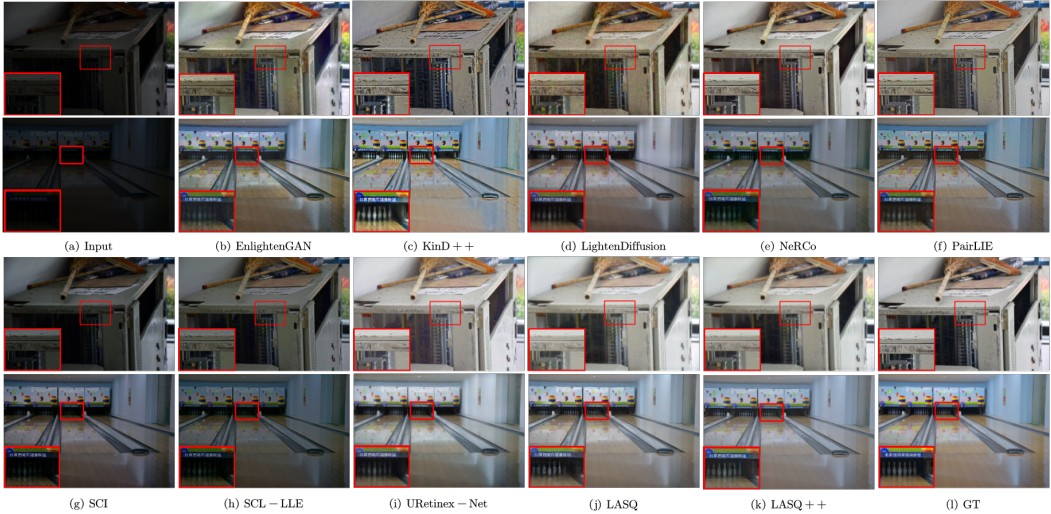

Figure 3: Qualitative comparison of our method and competitive methods on the LOLv1 and LSRW test sets. "LASQ++" denotes the incorporation of unpaired normal-light references.

## 4 Experiments

### 4.1 Experimental Settings

All experiments are carried out on a cluster of four NVIDIA A800 GPUs under Python 3.9 and PyTorch 2.0, with a fixed batch size of 16. We employ the Adam optimizer (30), setting the denoising diffusion process learning rate to $2 \times 10^{-5}$, while using a sampling ratio $k = 3$. The hyperparameters $\lambda_{\text{d}}$, $\lambda_{\text{g}}$ and $\lambda_{\text{GAN}}$ (if activated) are set to 0.9, 0.005 and 0.7 respectively. Noise estimation during diffusion training is performed using the U-Net (31) architecture with $T = 1000$ time steps.

### 4.2 Datasets and Metrics

We validated our approach using both paired and unpaired low-light benchmarks. For the paired evaluation, we used the LOLv1 (22) and LSRW (32) test sets—each comprising matched low- and normal-illumination image pairs—and reported restoration fidelity via PSNR and SSIM (33), alongside the full-reference perceptual score LPIPS (34). To assess performance in the absence of ground-truth references, we then tested on the unpaired LIME (35), DICM (36), NPE (37), and VV (38) collections, measuring perceptual quality with the no-reference NIQE (39) and PI (40) metrics. Our comparative study encompasses six supervised approaches (RetinexNet (22), KinD++ (23), LCDPNet (21), URetinexNet (41), SMG (17) and PyDiff (42)) and six unsupervised approaches (Zero-DCE (25), EnlightenGAN (24), SCI (19), PairLIE (43), SCL-LLE (44), LightenDiffusion (10) and NeRCo (18).

### 4.3 Qualitative Comparison

As shown in Fig. 3, LASQ achieves enhanced local brightness adaptation and superior detail fidelity comparable to supervised methods URetinexNet (41) and KinD++ (23) on ground truth-annotated datasets, while demonstrating improved domain-adaptive color reproduction through integration with unpaired normal-light images in LASQ++. By contrast, existing methods exhibit distinct limitations: SCI (19) and SCL-LLE (44) suffer from persistent underexposure, whereas EnlightenGAN (24) produces blurred structural details, and NeRCo (18) tends to generate localized over-exposure artifacts. Furthermore, Fig. 4 confirms LASQ's exceptional performance in real-world scenarios through its complete avoidance of local overexposure, noise amplification, and artifacts that persistently affect other methods: EnlightenGAN (24), NeRCo (18) and PairLIE (43) notably show severe localized overexposure and lens flare artifacts, and even supervised approaches URetinexNet (41) and KinD++ (23) still struggle to fully suppress these issues. Crucially, LASQ maintains natural

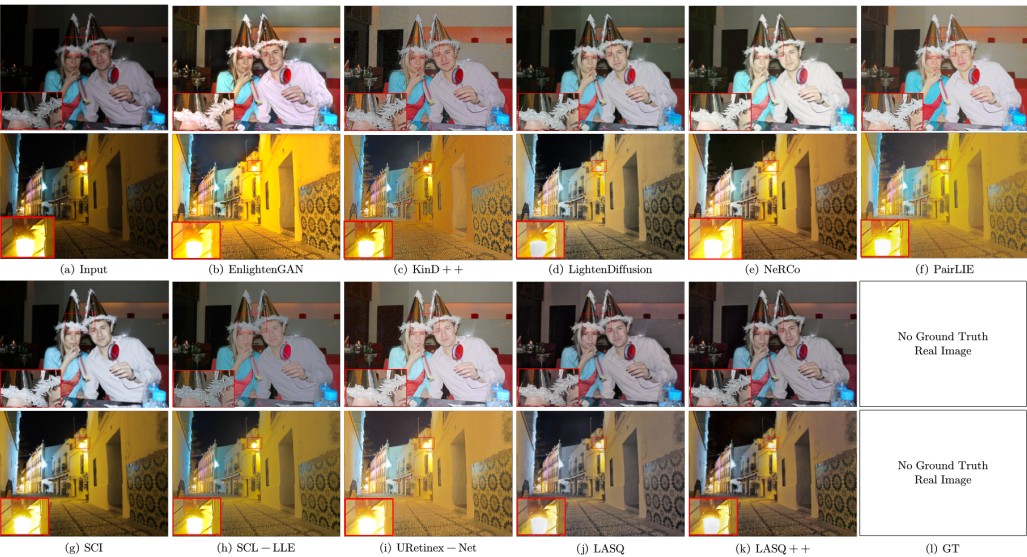

Figure 4: Qualitative comparison of our method and competitive methods on the LIME, and VV datasets. More results will be provided in the Appendix.

scene characteristics without compromising detail fidelity or color consistency, thereby demonstrating unprecedented cross-scenario generalization capability across both constrained laboratory settings and unconstrained environmental conditions. This comprehensive evaluation systematically validates LASQ's technical superiority in terms of adaptive illumination control, artifact suppression, and domain transfer effectiveness. More results will be provided in the Appendix.

## 4.4 Quantitative Comparison

The quantitative evaluation results across diverse datasets are summarized in Table 1, where LASQ demonstrates performance parity with leading supervised techniques on LOLv1 (22) and LSRW (32) while achieving state-of-the-art results among unsupervised methods through integration of unpaired normal-light images. Notably, on datasets DICM (36), NPE (37), and VV (38), LASQ outperforms existing approaches across most perceptual metrics, thereby confirming its intrinsic generalization prowess without domain-specific adaptation. Although normal-light reference integration improves color fidelity, its tendency toward overfitting partially counteracts the model's inherent generalization capacity, resulting in metric degradation in LASQ++. Crucially, this performance highlights LASQ's fundamental advantage in balancing domain adaptation with cross-scenario robustness, whereas LASQ++ prioritizes target-domain color accuracy at the expense of slight generalization capability. The extended experimental results, including comprehensive qualitative analyses and quantitative evaluations, are provided in the Appendix.

## 4.5 Computational Cost

We show the computational complexity metrics in Table 2 (NVIDIA A800, LOLv1 dataset). The coarse-to-fine MCMC sampling mechanism is only used during training and is embedded into the forward diffusion process. It guides the model to traverse luminance layers in a hierarchical manner, enabling structured learning of light propagation. During inference, our model only performs the denoising step conditioned on the low-light input within the diffusion model, which is significantly more efficient.

We compare LASQ against both early non-diffusion-based methods (e.g., EnlightenGAN, KinD++), and recent diffusion-based approaches (e.g., WCDM, LightenDiffusion). While the early methods are lightweight, their performance lags far behind diffusion-based models across all key metrics. Existing diffusion models, although significantly more effective, tend to suffer from high computational cost due to deep architectures and iterative sampling. LASQ, while maintaining the performance

Table 1: The quantitative comparison results of partial experiments, with the best-performing results marked in red and the second-best in blue. The notations "SL" and "UL" respectively represent supervised and unsupervised learning approaches. "LASQ++" denotes the incorporation of unpaired normal-light references

| Type | Method | LOLv1 | | | LSRW | | | DICM | | NPE | | VV | |
|---|---|---|---|---|---|---|---|---|---|---|---|---|---|
| | | PSNR↑ | SSIM↑ | LPIPS↓ | PSNR↑ | SSIM↑ | LPIPS↓ | NIQE↓ | PI↓ | NIQE↓ | PI↓ | NIQE↓ | PI↓ |
| SL | RetinexNet | 16.774 | 0.462 | 0.390 | 15.609 | 0.414 | 0.393 | 4.487 | 3.242 | 4.732 | 3.219 | 5.881 | 3.727 |
| | KinD++ | 17.752 | 0.758 | 0.198 | 16.085 | 0.394 | 0.366 | 4.027 | 3.999 | 4.005 | 3.144 | 3.586 | 2.773 |
| | LCDPNet | 14.506 | 0.575 | 0.312 | 15.689 | 0.474 | 0.344 | 4.110 | 3.250 | 4.126 | 3.127 | 5.039 | 3.347 |
| | URetinexNet | 19.842 | 0.824 | 0.128 | 18.271 | 0.518 | 0.295 | 4.774 | 3.565 | 4.028 | 3.153 | 3.851 | 2.891 |
| | SMG | 23.814 | 0.809 | 0.144 | 17.579 | 0.538 | 0.456 | 6.224 | 4.228 | 5.300 | 3.627 | 5.752 | 3.757 |
| | PyDiff | 23.275 | 0.859 | 0.108 | 17.264 | 0.510 | 0.335 | 4.499 | 3.792 | 4.082 | 3.268 | 4.360 | 3.678 |
| UL | Zero-DCE | 14.861 | 0.562 | 0.330 | 15.867 | 0.443 | 0.315 | 3.951 | 3.149 | 3.826 | 2.918 | 5.080 | 3.307 |
| | EnGAN | 17.606 | 0.653 | 0.319 | 17.106 | 0.463 | 0.322 | 3.832 | 3.256 | 3.775 | 2.953 | 3.689 | 2.749 |
| | SCI | 14.784 | 0.525 | 0.333 | 15.242 | 0.419 | 0.321 | 4.519 | 3.700 | 4.124 | 3.534 | 5.312 | 3.648 |
| | PairLIE | 19.514 | 0.731 | 0.254 | 17.602 | 0.501 | 0.323 | 4.282 | 3.469 | 4.661 | 3.543 | 3.373 | 2.734 |
| | SCL-LLE | 10.754 | 0.506 | 0.382 | 13.110 | 0.310 | 0.396 | 5.129 | 3.809 | 4.873 | 3.692 | 5.513 | 4.316 |
| | NeRCo | 19.738 | 0.740 | 0.239 | 17.844 | 0.535 | 0.371 | 4.107 | 3.345 | 3.902 | 3.037 | 3.765 | 3.094 |
| | LigDiff | 20.453 | 0.803 | 0.192 | 18.555 | 0.539 | 0.311 | 3.724 | 3.144 | 3.618 | 2.879 | 2.941 | 2.558 |
| | LASQ | 20.375 | 0.814 | 0.191 | 18.137 | 0.547 | 0.308 | 3.715 | 3.128 | 3.571 | 2.764 | 2.777 | 2.623 |
| | LASQ++ | 20.481 | 0.807 | 0.205 | 18.584 | 0.540 | 0.316 | 3.723 | 3.137 | 3.601 | 2.789 | 2.850 | 2.691 |

Table 2: Comparison of computational efficiency and resource usage.

| Method | FLOPs (G) | Params (M) | Inference Time (ms) | Memory Usage (MB) |
|---|---|---|---|---|
| EnlightenGAN | 16.45 | 8.64 | 70.16 | 241.48 |
| KinD++ | 17.49 | 8.27 | 4279.70 | 372.19 |
| NeRCo | 184.20 | 23.30 | 354.77 | 2320.87 |
| PairLIE | 81.84 | 0.34 | 900.70 | 3499.79 |
| SCI | 0.13 | - | 50.14 | 20.01 |
| SCL-LLE | 19.01 | 0.08 | 60.59 | 324.31 |
| URetinex | 81.35 | 0.34 | 129.70 | 568.48 |
| WCDM | 374.47 | 22.92 | 206.66 | 6017.86 |
| LightenDiffusion | 367.99 | 27.83 | 257.94 | 8049.95 |
| LASQ | 219.75 | 24.08 | 213.89 | 6496.68 |

advantages of diffusion models, achieves inference efficiency comparable to non-diffusion-based methods. This makes it highly suitable for real-world deployment. Considering the substantial performance gain without reliance on reference images, the moderate computational overhead of LASQ is a practical and acceptable trade-off.

## 5 Ablation

### 5.1 Fixed Luminance Adjustment

We replace our adaptive MCMC-based luminance adaptation with static, hand-crafted functions (e.g., global gamma correction or fixed tone curves) drawn from prior LLIE methods (45; 46; 47; 48). Embedding these into the forward diffusion degrades PSNR, SSIM, and LPIPS (Table 3), showing reduced feature richness and generative fidelity (Fig. 5). In contrast, our physics-driven, multi-scale operator—via power-law stratification and adaptive sampling—better balances global consistency and local detail, yielding stronger cross-scenario generalization and perceptual quality.

### 5.2 Limited Hierarchy

We keep adaptive MCMC sampling but limit the operator to two layers: a global adjustment and per-pixel correction, omitting mid-level power-law strata. This two-layer variant still beats the fixed baseline, yet its PSNR, SSIM, and perceptual metrics fall short of the full LASQ (Table 3), confirming that intermediate layers are essential for smoothly refining illumination and preserving both quantitative performance and visual fidelity.

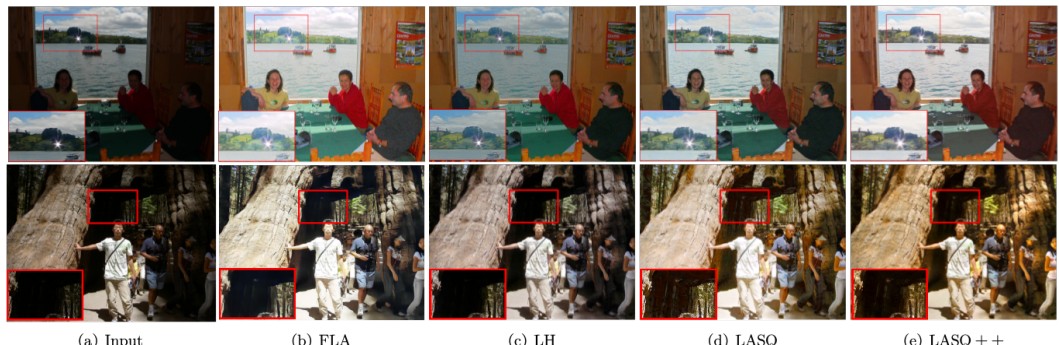

| (a) Input | (b) FLA | (c) LH | (d) LASQ | (e) LASQ++ |

Figure 5: The qualitative results of ablation studies.

Table 3: Quantitative results of ablation studies. The "FLA" and "LH" respectively represent the ablation of Fixed Luminance Adjustment and Limited Hierarchy.

| Method | LOLv1 | | | LSRW | | | DICM | | NPE | | VV | |
|---|---|---|---|---|---|---|---|---|---|---|---|---|
| | PSNR↑ | SSIM↑ | LPIPS↓ | PSNR↑ | SSIM↑ | LPIPS↓ | NIQE↓ | PI↓ | NIQE↓ | PI↓ | NIQE↓ | PI↓ |
| FLA | 16.741 | 0.715 | 0.273 | 15.490 | 0.508 | 0.399 | 4.265 | 3.529 | 3.937 | 3.114 | 3.683 | 3.007 |
| LH | 19.139 | 0.792 | 0.243 | 18.026 | 0.522 | 0.333 | 3.759 | 3.396 | 3.648 | 2.996 | 3.006 | 2.730 |
| LASQ | 20.375 | 0.814 | 0.191 | 18.137 | 0.547 | 0.308 | 3.715 | 3.128 | 3.571 | 2.764 | 2.777 | 2.623 |
| LASQ++ | 20.481 | 0.807 | 0.205 | 18.584 | 0.540 | 0.316 | 3.723 | 3.137 | 3.601 | 2.789 | 2.850 | 2.691 |

## 5.3 Hyperparameter Sensitivity

As illustrated in the table, we systematically varied key hyperparameters including $\alpha$, $\eta$, $\lambda_d$, and $\lambda_g$ over a range of values ( $\beta_{\mathcal{P}}$ is determined by $\eta$ ). The results show that while performance slightly fluctuates with different settings, the overall impact on metrics remains moderate. For instance, varying between 0.05 and 0.6 only causes a minor PSNR change (within 0.3 dB) and negligible shifts in perceptual scores. Similarly, other hyperparameters demonstrate a stable trend without sharp degradation. These experiments demonstrate that our method is not overly sensitive to hyperparameter selection, and maintains consistently strong performance across a broad range of settings—highlighting its robustness, practical stability, and generalization potential.

Table 4: Ablation study on key hyperparameters. The best results for are highlighted in bold.

| Param | Value | PSNR↑ | LPIPS↓ | SSIM↑ |
|---|---|---|---|---|
| $\alpha$ | 0.05 / 0.15 / 0.3 / 0.6 | 17.81 / **18.10** / 17.92 / 17.84 | **0.319** / 0.322 / 0.320 / 0.324 | 0.512 / **0.543** / 0.530 / 0.519 |
| $\eta$ | 0.1 / 1.0 / 3.0 / 6.0 | 17.85 / **18.35** / 18.17 / 17.95 | 0.335 / **0.321** / 0.324 / 0.329 | 0.537 / 0.543 / **0.546** / 0.540 |
| $\lambda_d$ | 0.1 / 1.0 / 10 / 20 | 17.82 / **18.04** / 17.85 / 17.87 | 0.324 / **0.315** / 0.318 / 0.323 | 0.545 / **0.553** / 0.549 / 0.531 |
| $\lambda_g$ | 0.001 / 0.005 / 0.01 / 0.1 | 17.76 / **18.22** / 18.16 / 17.88 | 0.312 / 0.310 / **0.309** / 0.311 | 0.536 / **0.548** / 0.547 / 0.540 |

## 6 Conclusion

The proposed LASQ framework reorients the LLIE challenge by merging illumination continuity physics with deep-learning, moving beyond conventional pixel-level methods. It redefines LLIE as a continuous stochastic task using adaptive MCMC, capturing real illumination through layered luminance analysis. Transitioning from reliance on paired data to unsupervised layer exploration, it improves generalization and can support reference contexts. It balances overall illumination with local detail, effectively resolving fidelity vs. adaptability issues, while avoiding gamma-related biases and overfitting of supervised methods. These advancements underscore a broader insight: LLIE must evolve from pixel-wise function approximation to spatiotemporal reconstruction of light dynamics. Future work should explore dynamic power-law parameterization for time-varying scenes and hardware-software co-design to align statistical priors with sensor-specific noise profiles. Narrowing these gaps will enhance computational imaging systems to better mimic biological vision in low light.

## Acknowledgments and Disclosure of Funding

This work is supported by the National Natural Science Foundation of China under Grant 62576350, 62131020, 62376283 and 62531026.

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

# A    The Detailed LASQ Workflow

**Luminance Variation Coordinate System:**  To systematically characterize low-light degradation, we establish the Luminance Variation (LV) coordinate system that geometrically encodes pixel-wise illumination relationships. Let $\mathcal{I}_L \in [0,1]^{H \times W}$ and $\mathcal{I}_N \in [0,1]^{H \times W}$ denote paired low/normal-light images. For each pixel $i$, we define its luminance state as:

$$\left\{ s_i = \left( \mathcal{I}_L^{(i)}, \mathcal{I}_N^{(i)} \right) \right\}_{i=1}^{I}. \tag{13}$$

**Power Law like Transformation:** As visualized in Fig. 6 (a), these coordinate points $s_i$ constitute a geometric manifold where the horizontal/vertical axes respectively represent low/normal-light intensity spaces. This formulation provides two critical physical insights: 1) Each $s_i$ encodes a pixel-specific luminance attenuation pattern; 2) The points' spatial distribution reflects global illumination degradation statistics. The transition from Fig. 6 (b) to (d) reveals our hierarchical modeling strategy. For an individual $s_i$, we derive its local transformation strategy through:

$$y = ax^{\kappa}, x \in [0,1], \kappa > 0, a > 0, \tag{14}$$

where $\kappa$ quantifies the exposure compensation magnitude at pixel $i$ (Fig. 6 (b)). We note that $\kappa$ exhibits spatial correlation with scene content—lower $\kappa$ values (steeper curves) correspond to regions requiring aggressive enhancement (e.g., shadows), while higher $\kappa$ preserves highlights.

Extending to multiple pixels (Fig. 6(c)), we observe heterogeneous curve families governed by parameter distributions $\{\kappa_p\}$. Each curve represents a distinct luminance adjustment strategy:

- Highlight preservation ($\kappa_p \to 1$): Linear mapping maintaining high-intensity features.
- Mid-tone enhancement ($0.5 < \kappa_p < 1$): Concave curves amplifying moderate intensities.
- Shadow recovery ($\kappa_p < 0.5$): Convex curves dramatically boosting dark regions.

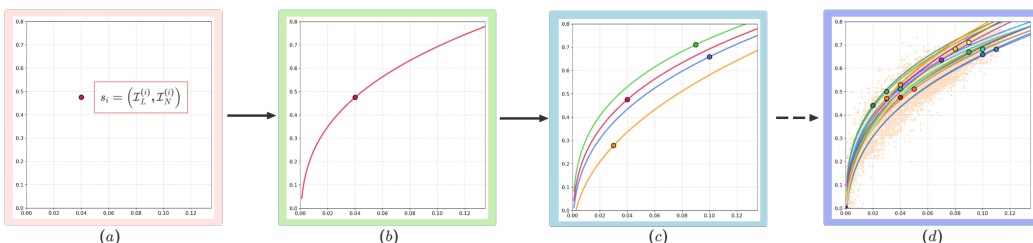

(a)          (b)          (c)          (d)

Figure 6: Luminance Variation Coordinate and the progressive modeling on full-domain luminance intensity elements via power law curves. (a) Luminance variation coordinate with pink background, where each sample $s_i$ is geometrically determined by the horizontal low-light intensity $\mathcal{I}_L^{(i)}$ and vertical normal-light intensity $\mathcal{I}_N^{(i)}$, marked by a central red dot; (b) Single-sample modeling (green background) demonstrating that the luminance transformation of individual elements follows a red-colored power law curve; (c) Multi-sample extension (cyan background) showing diversified mapping relationships through color-coded power law curves corresponding to different elements; (d) Full-domain fitting (purple background) achieved by dense overlapping curves that comprehensively cover the coordinate space.

Full-domain fitting in Fig. 6 (d) employs dense overlapping curves to approximate the entire $\{s_i\}_{i=1}^{I}$ distribution. While achieving theoretical completeness through

$$\bigcup_{i=1}^{I} \{(x, ax^{\kappa_i}) \,|\, x \in [0,1]\} \to \text{LV coordinate coverage}, \tag{15}$$

this over-parameterized paradigm introduces critical limitations. First, optimizing curves at the pixel level ignores spatial coherence, leading to artifacts as adjacent pixels apply divergent enhancement

strategies. Second, estimating $\kappa_p$ becomes problematic in low-intensity areas ($\mathcal{I}_L^{(i)} < 0.1$), where slight intensity shifts cause significant curve fluctuations. Lastly, focusing heavily on the individual $\{\kappa_p\}$ parameters greatly restricts practical implementation due to poor generalization. Therefore, we statistically formulate an MCMC-sampling-based physics-aware $\{\kappa_p\}$ prediction to preserves the LV system's statistical advantages while mitigating overfitting.

**Hierarchical Sampling of Luminance Adaptation Operators:** Given a low-light RGB image $\mathcal{I}_L \in \mathbb{R}^{H \times W \times 3}$, we first convert it into the YUV color space and perform guided filtering to compute the smoothed illumination map $G(i, j)$, where $(i, j)$ denotes pixel coordinates. The local linear coefficients in the guided filtering process can be formulated as:

$$a(i, j) = \frac{\sigma_L^2(i, j)}{\sigma_L^2(i, j) + \epsilon}, \quad b(i, j) = \mu_L(i, j) - a(i, j) \cdot \mu_L(i, j), \tag{16}$$

where $\mu_L(i, j)$ represents the local window mean and $\sigma_L^2(i, j)$ denotes the local window variance. Through coefficient smoothing, $G(i, j)$ is obtained as:

$$G(i, j) = \bar{a}(i, j) \cdot L(i, j) + \bar{b}(i, j), \tag{17}$$

where $\bar{a}$ and $\bar{b}$ denote the mean-filtered results of $a$ and $b$, respectively. This luminance map $G(i, j)$ is then aggregated over arbitrary image regions $\mathcal{P} \subseteq [1, H] \times [1, W]$ to obtain the regional scalar $G_\mathcal{P}$, which quantifies local luminance characteristics and serves as the input to compute the LAO $\gamma_\mathcal{P}$ via the parameterized transformation introduced in the main text **Section 3.2 Statistical Sampling Process**.

**Coarse-to-Fine Hierarchy Gaussian distribution** As illustrated in Fig.7, LAOs are sampled from a truncated Gaussian distribution $\mathcal{N}_{\text{trunc}}\left(\mu = \gamma_0, \sigma^2; \gamma_{\min}, \gamma_{\max}\right)$. Commencing from a globally balanced state, spatially-varying layers are iteratively introduced to simulate the continuity of real-world luminance transformations. The sampling operator employs a truncated Gaussian distribution wherein high-probability candidates correspond to physically plausible global adaptations, while low-probability candidates represent local refinements. This characteristic inherently reflects the natural illumination transitions where extreme pixel-level modifications rarely occur. The transition between states is governed by a kernel that ensures smooth evolution across iterations:

$$q\left(\gamma_{\mathcal{P},z}^{(n)} \mid \gamma_{\mathcal{P},z-1}^{(n)}\right) = \mathcal{N}_{\text{trunc}}\left(\gamma_{\mathcal{P},z}^{(n)} \mid \gamma_{\mathcal{P},z-1}^{(n)}, \lambda^2; \gamma_{\min}, \gamma_{\max}\right), \tag{18}$$

This hierarchical sampling yields a sequence of LAO sets $\Gamma = \{\Gamma_n\}_{n=1}^N$, with each $\Gamma_n = \left\{\gamma_{\mathcal{P},z}^{(n)}\right\}_{z=1}^{2^{n-1}}$ representing the spatially adaptive enhancement parameters. When applied, these sets give rise to a family of luminance-enhanced images $\mathcal{H} = \{\mathcal{I}_H^{(n)}\}_{n=1}^N$, capturing a coarse-to-fine progression of illumination correction. This structure enables flexible exploration of global and local enhancement effects through state-dependent sampling transitions across the luminance field.

**Hierarchically-Guided Diffusion:** The forward process employs a hierarchically-guided diffusion framework to progressively enhance low-light images through semantically-aligned noise injection. By encoding the input $\mathcal{I}_L$ and its luminance-enhanced variants $\mathcal{H}$ into latent features $\mathcal{F}_L$ and $\left\{\mathcal{F}_H^{(n)}\right\}_{n=1}^N$, the method establishes temporal correspondence between diffusion steps and multi-scale guidance via a mapping function $\psi(t)$. This alignment dynamically steers the noise trajectory, where each transition step incorporates hierarchical semantic shifts derived from $\mathcal{F}_H^{(\psi(t))}$ to modulate the degradation path. Throughout the forward pass, coarse-to-fine illumination characteristics are gradually imprinted into the diffusion process. During reverse denoising, the network $\epsilon_\theta$ iteratively removes noise while synergizing hierarchical cues from original low-light features $\mathcal{F}_L$. The denoising trajectory is regularized by a dual objective: a noise prediction loss ensures faithful reconstruction aligned with hierarchical semantics, while a global constraint $\mathcal{L}_g$ enforces the structural consistency.

**Adversarial Discriminator for Iterative Refinement:** To optionally further enhance realism and perceptual fidelity—only when a normal-light reference is available—we introduce a discriminator $\mathcal{D}_\phi$ alongside our diffusion-based generator $G_\theta$. Given an unpaired sample $\mathcal{I}_{\text{normal}} \sim p_{\text{normal}}$ (if available) from the normal-light distribution, the discriminator learns to distinguish between true normal-light images and our enhanced outputs $\hat{\mathcal{I}}_N = G_\theta(\mathcal{I}_L)$. We therefore cast the overall framework as a hybrid

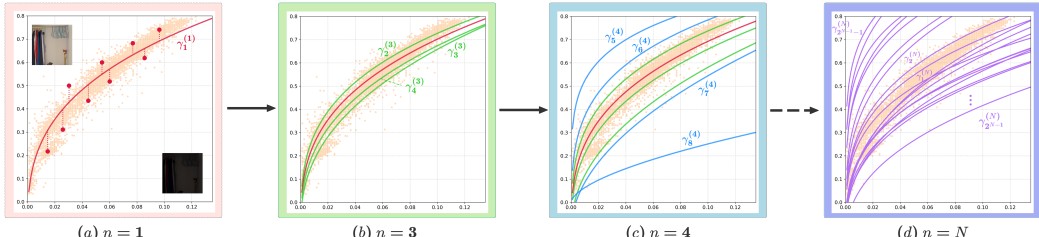

$(a)\ n = 1$      $(b)\ n = 3$      $(c)\ n = 4$      $(d)\ n = N$

Figure 7: The hierarchical MCMC sampling scheme to generate LAO sets. (a) Initial state ($n = 1$) with a uniform LAO operator $\gamma_1^{(1)}$ (red curve) globally adjusting pixel intensities via orthogonal projections to fit shallow orange sample points (approximate power-law distribution); (b) Intermediate sampling ($n = 3$) generates three green curves ($\gamma_{i=2,3,4}^{(3)}$) near the red curve, grounding on the Markov chain posterior; (c) Increased sampling times ($n = 4$) introduces external blue curves ($\gamma_{i=5,6,7,8}^{(4)}$) , which entail power law functions ($\gamma_{i=5,8}^{(4)}$) that are outlying of the main distribution area; (d) Converged state ($n = N$) achieves pixel-level granularity through dense overlapping curves, dynamically fitting the orange distribution.

diffusion–GAN, optimizing both the variational diffusion objective and an adversarial objective in an alternating fashion.

Concretely, at each training iteration $t$, we perform the discriminator update as:

$$\phi \leftarrow \phi - \eta_\phi \, \nabla_\phi \, \mathcal{L}_{\mathrm{adv}}(\mathcal{D}_\phi; G_\theta),$$
$$\mathcal{L}_{\mathrm{adv}} = -\mathbb{E}_{\mathcal{I}_{\mathrm{normal}} \sim p_{\mathrm{normal}}}\big[\log \mathcal{D}_\phi(\mathcal{I}_{\mathrm{normal}})\big] - \mathbb{E}_{\mathcal{I}_L}\big[\log\big(1 - \mathcal{D}_\phi(G_\theta(\mathcal{I}_L))\big)\big]; \tag{19}$$

and the generator update can be formulate as:

$$\theta \leftarrow \theta - \eta_\theta \, \nabla_\theta \big(\mathcal{L}_{\mathrm{total}} + \lambda_{\mathrm{GAN}} \, \mathcal{L}_{\mathrm{G\text{-}adv}}\big),$$
$$\mathcal{L}_{\mathrm{G\text{-}adv}} = -\mathbb{E}_{\mathcal{I}_L}\big[\log \mathcal{D}_\phi(G_\theta(\mathcal{I}_L))\big], \tag{20}$$

where $\eta_\phi, \eta_\theta$ are learning rates, and $\lambda_{\mathrm{GAN}}$ balances adversarial and diffusion losses. Therefore, the full generator loss becomes:

$$\mathcal{L}_{\mathrm{total}} = \lambda_d \mathcal{L}_d + \lambda_g \mathcal{L}_g + \lambda_{\mathrm{GAN}} \underbrace{\mathbb{E}_{\mathcal{I}_L}\big[-\log \mathcal{D}_\phi(G_\theta(\mathcal{I}_L))\big]}_{\text{adversarial penalty}}. \tag{21}$$

By unifying physically grounded diffusion priors with adversarial discrimination, our hybrid framework yields outputs that not only preserve structural details and smooth lighting transitions but also exhibit the sharp textures and natural contrasts characteristic of true normal-light images.

**Inference Stage:** During the inference phase, the model synthesizes an optimal enhanced representation $\hat{\mathcal{F}}_N$ that balances global illumination correction with local texture fidelity through implicit latent sampling guided by $\mathcal{F}_L$, starting from Gaussian random noise. This representation is subsequently decoded into the final output $\hat{\mathcal{I}}_N$ through the learned mapping function.

## B    The Algorithmic Framework

Algorithm 1 summarizes the full LASQ (Luminance-Adaptation Sampling and Hierarchically-Guided Diffusion) workflow. It first extracts a latent representation via an encoder, then performs multi-scale luminance-adaptive sampling to generate a hierarchy of intermediate images. These are used to guide both the forward and reverse diffusion processes, and finally a decoder reconstructs the enhanced high-quality output, optionally applying adversarial training for further refinement.

**Algorithm 1** The LASQ pipeline.

---

1: **Input:** Low-light image $\mathcal{I}_L$

2: **Hierarchical LAO sampling:**

3:  Initialize $\Gamma = \{\}, \mathcal{I}_H^{(0)} = \{\}$

4:  Compute mean luminance $G_p$, then

5:   $\gamma_{\mathcal{P}} = (\alpha + G_{\mathcal{P}})^{\beta_{\mathcal{P}}}$

6:  **for** $n = 1 \dots N$ **do**

7:   Sample $\gamma \sim \mathcal{N}_{\text{trunc}}(\gamma_0, \sigma^2, \gamma_{\min}, \gamma_{\max})$

8:   $\Gamma_n = \left\{ \gamma_{\mathcal{P},z}^{(n)} \right\}_{z=1}^{2^{n-1}}$ ; Update $I_H^{(n)}$

9:  **end for**

10:  Collect $\mathcal{H} = \{\mathcal{I}_H^{(n)}\}_{n=1}^N$

11: **Hierarchically-Guided diffusion:**

12:  $\mathcal{F}_L \leftarrow \mathcal{E}(\mathcal{I}_L); \left\{ \mathcal{F}_H^{(n)} \right\}_{n=1}^N \leftarrow \mathcal{E}(\mathcal{H})$

13:  Define $\psi(t) = \lceil tN/T \rceil$, set $x_0 = \mathcal{F}_H^{(0)}$

14:  **while** not converged **do**

15:   **for** $t = 1 \dots T$ **do**

16:    $x_0 = \mathcal{F}_H^{(\psi(t))}$

17:    $x_t \sim \mathcal{N}(\sqrt{1 - \beta_t} x_{t-1}, \beta_t I)$

18:    Perform gradient descent steps on $\nabla_\theta \| \epsilon - \epsilon_\theta (x_t, t, \mathcal{F}_L) \|^2$

19:   **end for**

20:   **for** $t = T \dots 1$ **do**

21:    Predict $\epsilon_\theta(\hat{x}_t, t, \mathcal{F}_L)$

22:    $\hat{x}_{t-1} = \frac{1}{\sqrt{1-\beta_t}}(\hat{x}_t - \beta_t \epsilon_\theta) + \sigma_t b$

23:   **end for**

24:   $\hat{\mathcal{I}}_N = \mathcal{D}(\hat{x}_0)$

25:   Perform gradient descent steps on $\nabla_\theta \left\| \hat{\mathcal{I}}_N - \mathcal{D}(\mathcal{F}_H^{(\psi(0))}) \right\|_1$

26:   **Optional:** $\nabla_\theta \left[ -\log \mathcal{D}_\phi(G_\theta(\mathcal{I}_L)) \right]; \nabla_\phi \left[ \log \mathcal{D}_\phi (\mathcal{I}_{\text{normal}}) + \log (1 - \mathcal{D}_\phi (G_\theta (\mathcal{I}_L))) \right]$

27:  **end while**

28: **Return** $\hat{\mathcal{I}}_N$

---

# C Supplementary Experiments and Extended Results

**Visual Comparison:** As demonstrated in the supplemental visual comparisons (Fig. 8-10), LASQ exhibits remarkable stability across both controlled laboratory settings and challenging real-world scenarios. Figures 8 and 9 systematically visualize LASQ's consistency in ground truth-annotated scenes, where it maintains precise alignment with reference images in terms of color temperature, dynamic range distribution, and fine-grained texture preservation. The visual trajectories across multiple test cases confirm that LASQ effectively decouples scene semantics from lighting interference, avoiding the common pitfalls of oversaturation or detail loss observed in supervised baselines. Fig. 10 further underscores LASQ's robustness under extreme conditions, specifically in nocturnal environments and localized overexposure scenarios—settings that are notoriously difficult for conventional methods. In night-time scenes with extremely low ambient illumination, LASQ demonstrates a pronounced ability to preserve structural integrity and suppress noise amplification, resulting in outputs that maintain visual coherence and readability. Unlike comparative methods that often introduce flare artifacts, chromatic aberrations, or abrupt luminance shifts under high-contrast lighting, LASQ adaptively regulates brightness and minimizes highlight clipping, ensuring that no important spatial detail is lost. Particularly in overexposed regions—such as headlights, reflective surfaces, or artificially illuminated zones—LASQ retains nuanced intensity gradients and avoids flattening or unnatural glow, thereby delivering smooth tonal transitions that remain faithful to human perceptual expectations. These results confirm that LASQ not only generalizes well across diverse lighting conditions but also excels where existing techniques tend to break down.

**Quantitative Results:** To provide a more comprehensive evaluation of LASQ and LASQ++ under realistic, unconstrained conditions, we present in Table 5 a quantitative comparison on five widely used no-reference low-light image enhancement benchmarks: DICM, NPE, VV, LIME, and MEF.

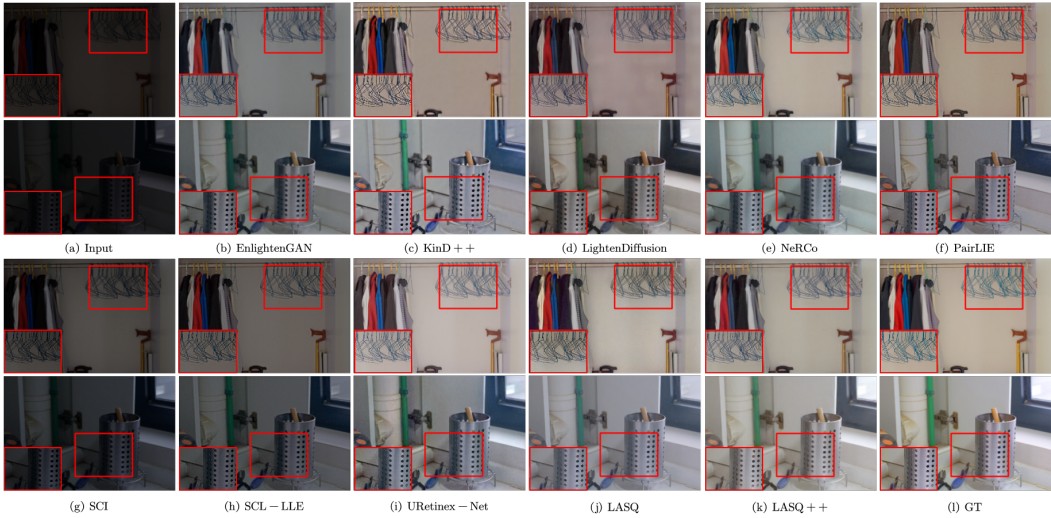

Figure 8: The supplemental qualitative comparison of our method and competitive methods on the LOLv1 test sets. "LASQ++" denotes the incorporation of unpaired normal-light references.

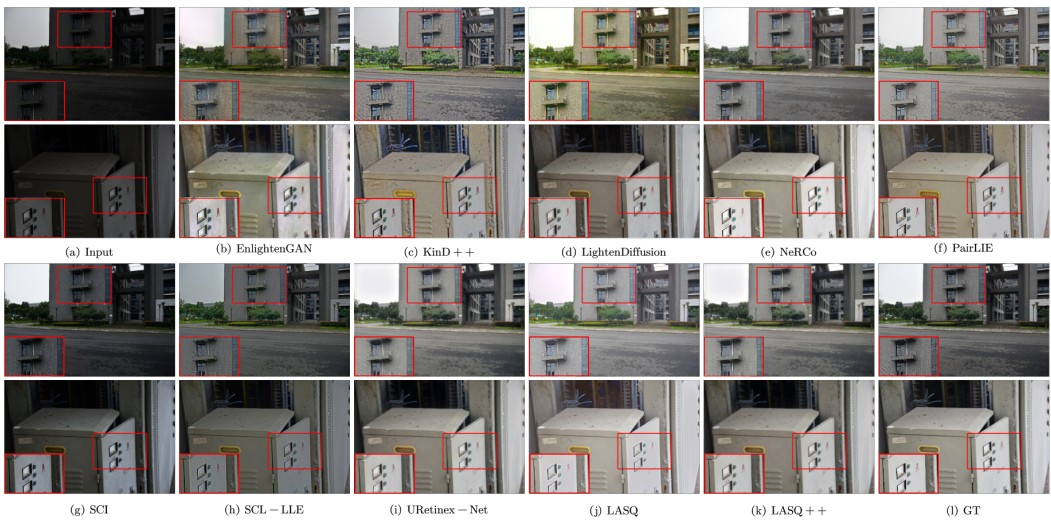

Figure 9: The supplemental qualitative comparison of our method and competitive methods on the LSRW test sets. "LASQ++" denotes the incorporation of unpaired normal-light references.

These datasets lack paired ground-truth (GT) normal-light references, making them a robust testbed for assessing generalization and real-world applicability. We evaluate performance using two standard no-reference metrics—NIQE and PI—and compare LASQ and LASQ++ against a broad range of state-of-the-art supervised (SL) and unsupervised (UL) methods. As shown, LASQ consistently achieves the best or second-best results across almost all datasets and metrics, outperforming all other unsupervised methods and remaining competitive with even several fully supervised approaches. This highlights the strong generalization capability of LASQ, which requires no paired data and yet adapts effectively to diverse lighting conditions. LASQ++, which incorporates unpaired normal-light references during training, pushes perceptual quality even further. While this slightly reduces robustness in the most challenging cross-domain settings, it reinforces the complementary strengths of our two models: LASQ is ideal for broad deployment due to its high robustness and adaptability, whereas LASQ++ excels when enhanced fidelity is needed in target-specific domains.

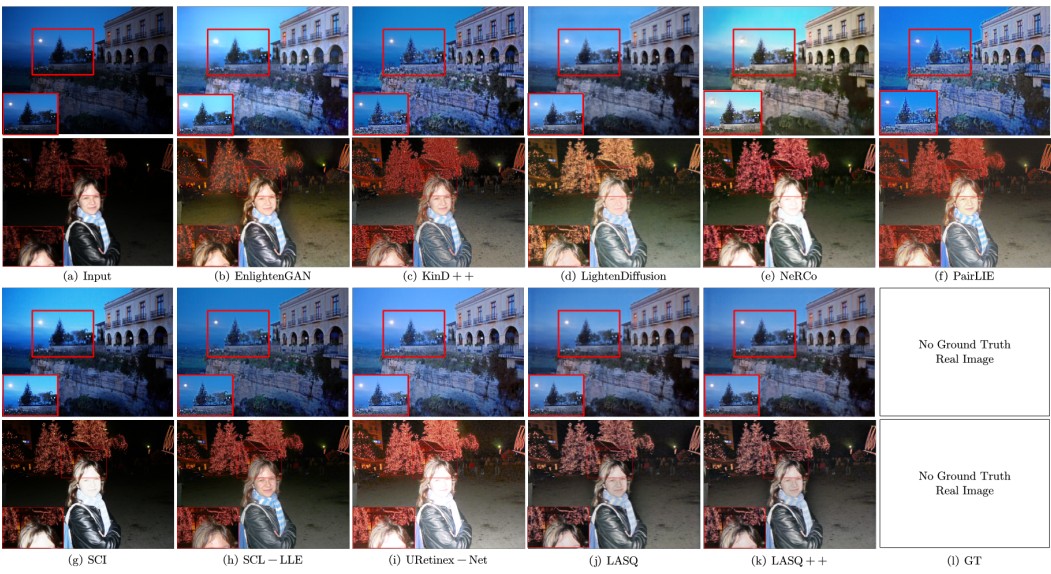

Figure 10: Visual method comparison in challenging scenarios. "LASQ++" denotes the incorporation of unpaired normal-light references.

## D   Computation Resources and Hyper-parameter Tuning

All experiments were conducted on a high-performance computing node equipped with four NVIDIA A100 80GB GPUs interconnected via NVLink. The system specifications are as follows:

- **OS**: Ubuntu 22.04 LTS with Linux 5.15 kernel

- **CPU**: Dual AMD EPYC 7763 64-Core @ 2.45GHz (128 cores/256 threads)

- **GPU Interconnect**: NVLink 3.0 (600GB/s bisectional bandwidth)

- **Memory**: 1TB DDR4 ECC @ 3200MHz

- **Storage**: 16TB NVMe SSD RAID (3.5GB/s sustained read)

- **Accelerators**: 4×NVIDIA A100 80GB (FP32: 19.5 TFLOPS, FP16: 312 TFLOPS)

The hyperparameter configuration for the proposed framework is comprehensively summarized as follows. All experiments are conducted on four NVIDIA A800 GPUs utilizing Python 3.9 and PyTorch 2.0 with a fixed batch size of 16, employing the Adam optimizer with a denoising diffusion learning rate of $2 \times 10^{-5}$ and a sampling ratio $k = 3$. The loss weighting coefficients $\lambda_\mathrm{d}$, $\lambda_\mathrm{g}$, and $\lambda_\mathrm{GAN}$ are empirically set to 0.9, 0.005, and 0.7 respectively when adversarial training is activated. The diffusion process operates over T=1000 time steps with a U-Net-based noise estimation architecture. In the luminance adaptation framework, the power-law adjustment parameters $\alpha$, $\eta$, and $\delta$ governing local contrast enhancement are initialized to 2, 0.1, and 0.01 respectively, while the MCMC sampling process employs an adaptive step size $\lambda = 0.2$ to balance exploration-exploitation dynamics across hierarchy levels. The temporal mapping function $\psi(t)$ synchronizes $N = 100$ hierarchical guidance levels with the $T$-step diffusion through linear interpolation. The truncated Gaussian distribution for LAO sampling is bounded by $\gamma_\mathrm{min}$ and $\gamma_\mathrm{max}$ derived dynamically from image statistics, ensuring physically plausible luminance adjustments. For adversarial training augmentation, the discriminator $D_\phi$ maintains architectural hyperparameters aligned with LSGAN conventions (including convolutional layer configurations and spectral normalization usage), though implemented with standard binary cross-entropy objectives rather than least-squares formulations. The network depth and feature channel dimensions are adaptively scaled according to the spatial resolution of input pairs from the generator's decomposition process, while preserving LSGAN's fundamental design principles in layer progression and discriminator receptive field structure.

Table 5: Supplementary quantitative comparisons on the no-reference image datasets from the main text, with the best-performing results marked in red and the second-best in blue. The notations "SL" and "UL" respectively represent supervised and unsupervised learning approaches. "LASQ++" denotes the incorporation of unpaired normal-light references.

| Type | Method | DICM NIQE↓ | DICM PI↓ | NPE NIQE↓ | NPE PI↓ | VV NIQE↓ | VV PI↓ | LIME NIQE↓ | LIME PI↓ | MEF NIQE↓ | MEF PI↓ |
|---|---|---|---|---|---|---|---|---|---|---|---|
| SL | RetinexNet | 4.487 | 3.242 | 4.732 | 3.219 | 5.881 | 3.727 | 4.802 | 3.522 | 4.152 | 3.411 |
| | KinD++ | 4.027 | 3.999 | 4.005 | 3.144 | 3.586 | 2.773 | 4.035 | 3.217 | 3.874 | 3.285 |
| | LCDPNet | 4.110 | 3.250 | 4.126 | 3.127 | 5.039 | 3.347 | 4.128 | 3.332 | 3.912 | 3.398 |
| | URetinexNet | 4.774 | 3.565 | 4.028 | 3.153 | 3.851 | 2.891 | 3.987 | 3.104 | 3.721 | 3.185 |
| | SMG | 6.224 | 4.228 | 5.300 | 3.627 | 5.752 | 3.757 | 5.312 | 3.615 | 5.028 | 3.804 |
| | PyDiff | 4.499 | 3.792 | 4.082 | 3.268 | 4.360 | 3.678 | 4.412 | 3.685 | 4.228 | 3.572 |
| UL | Zero-DCE | 3.951 | 3.149 | 3.826 | 2.918 | 5.080 | 3.307 | 3.625 | 3.512 | 3.608 | 3.217 |
| | EnlightenGAN | 3.832 | 3.256 | 3.775 | 2.953 | 3.689 | 2.749 | 3.427 | 3.424 | 3.524 | 3.108 |
| | SCI | 4.519 | 3.700 | 4.124 | 3.534 | 5.312 | 3.648 | 4.032 | 3.518 | 3.892 | 3.415 |
| | PairLIE | 4.282 | 3.469 | 4.661 | 3.543 | 3.373 | 2.734 | 3.782 | 3.215 | 3.412 | 3.028 |
| | SCL-LLE | 5.129 | 3.809 | 4.873 | 3.692 | 5.513 | 4.316 | 5.104 | 4.302 | 4.872 | 4.115 |
| | NeRCo | 4.107 | 3.345 | 3.902 | 3.037 | 3.765 | 3.094 | 3.712 | 3.078 | 3.328 | 3.112 |
| | LightenDiffusion | 3.724 | 3.144 | 3.618 | 2.879 | 2.941 | 2.558 | 3.218 | 3.128 | 3.305 | 3.024 |
| | LASQ | 3.715 | 3.128 | 3.571 | 2.764 | 2.777 | 2.623 | 3.152 | 3.002 | 3.294 | 3.001 |
| | LASQ++ | 3.723 | 3.137 | 3.601 | 2.789 | 2.850 | 2.691 | 3.167 | 3.046 | 3.309 | 3.013 |

Table 6: Supplementary ablation results.

| Method | Pre-trained Dataset LOLv1 | LSRW | MEF | VV NIQE↓ | VV PI↓ | MEF NIQE↓ | MEF PI↓ | NPE NIQE↓ | NPE PI↓ | DICM NIQE↓ | DICM PI↓ | LIME NIQE↓ | LIME PI↓ |
|---|---|---|---|---|---|---|---|---|---|---|---|---|---|
| LASQ | ✓ | | | 2.777 | 2.623 | 3.294 | 3.001 | 3.571 | 2.764 | 3.715 | 3.128 | 3.152 | 3.002 |
| | | ✓ | | 2.801 | 2.637 | 3.307 | 3.019 | 3.584 | 2.762 | 3.750 | 3.160 | 3.191 | 3.222 |
| | | | ✓ | 2.882 | 2.703 | 3.287 | 3.010 | 3.627 | 2.758 | 3.744 | 3.177 | 3.248 | 3.335 |

# E  Ablation

As shown in Table 6, we present extend ablation studies to systematically evaluate the proposed framework. LASQ is trained on three datasets (LOLv1, LSRW, and MEF with manually curated training splits augmented via random cropping), followed by cross-dataset evaluations on MEF and VV benchmarks. Empirical results demonstrate LASQ's superior generalization capability in unseen scenarios, achieving consistent reconstruction quality under diverse illumination conditions. This evidence confirms that LASQ avoids overfitting to region-specific normal-light patterns and exhibits reduced sensitivity to pretraining data selection. Two principal advantages are emphasized: (1) Cross-domain adaptation through layered luminance modeling enables high-fidelity visual outputs without paired supervision; (2) Effective suppression of noise and artifacts under extreme low-light conditions, substantially improving perceptual quality. These advancements highlight LASQ's potential to redefine low-light image enhancement paradigms by harmonizing physical priors with data-driven learning.

# F  Limitations

The main limitations of our approach stem from its hierarchical sampling depth $N$ and diffusion step count $T$: as $N$ grows, the number of local correction patches doubles at each level, and as $T$ increases, each diffusion step must incorporate the corresponding hierarchical features, so jointly large $N$ and $T$ lead to exponential increases in computation and memory demands. Moreover, the final enhancement quality is quite sensitive to the choice of $N$ (too few layers underfits, too many layers overfits and amplifies noise) and $T$ (insufficient steps yield coarse results, excessive steps waste resources), as well as to key hyperparameters in the $\gamma$-calculation (e.g. the contrast gain terms $\eta$ and $\delta$ ), which must be manually tuned to balance enhancement strength against artifact suppression across different scenes.

## G   Impacts

Our LASQ advances low-light imaging for critical applications like target detection, self-driving and medical diagnostics, where reliable scene interpretation under poor illumination is essential. Our LASQ mimics natural light adaptation to improve nighttime safety. Its unsupervised design removes the need for paired data, expanding access to low-light enhancement for applications lacking annotated datasets.

