# OpenReview forum: "Luminance-Aware Statistical Quantization: Unsupervised Hierarchical Learning for Illumination Enhancement"
_NeurIPS.cc/2025/Conference — NeurIPS 2025 poster_

### Official Review · Reviewer_S427 · 2025-06-01

**Clarity:** 2
**Significance:** 2
**Originality:** 3
**Rating:** 4
**Confidence:** 5

**Summary:**

The paper presents Luminance-Aware Statistical Quantization (LASQ), an unsupervised framework for low-light image enhancement that treats illumination correction as a statistical sampling task rather than a direct pixel mapping. LASQ first shows, through empirical analysis, that natural low-to-normal luminance transitions follow power-law density curves; it then builds hierarchical “luminance adaptation operators” by sampling these curves with an adaptive Markov chain Monte Carlo strategy. These operators guide the forward path of a diffusion model, letting the network learn global-to-local corrections without paired supervision, while an optional adversarial head can exploit normal-light references when they exist.

**Questions:**

1. The framework relies on accurate local luminance statistics, heavy sensor noise or clipping might break the power-law assumption. Has the author considered this question?

2. Can the author provide a more adequate ablation experiment to verify the impact of the proposed method and the selection of hyperparameters on the results?

3. The optional adversarial head is said to improve texture fidelity when paired references are available; how much benefit does it add in strictly unpaired training, and does it introduce any instability during optimization?

4. The paper shows that LASQ slightly lags supervised diffusion models on PSNR for the LOL dataset; have you identified specific error patterns that explain this gap, and how might they be addressed?

I will adjust my score as appropriate according to the author's reply.
5. Could you provide failure examples involving extreme dynamic range, motion blur, or LED flicker, and discuss whether the current framework can be adapted to handle such cases?

**Ethical Concerns:**

["NO or VERY MINOR ethics concerns only"]

**Final Justification:**

I have read the authors’ response as well as the concerns raised by most of the reviewers. The explanations provided have given me greater confidence in the effectiveness of the proposed method, and I have adjusted my score accordingly. I also recommend that the authors further improve the figures and descriptions in future versions, and use vector graphics to enhance readability.

**Limitations:**

Yes

**Quality:**

2

**Strengths And Weaknesses:**

Strengths:

1. The method is rooted in a measurable physical regularity, the power-law distribution of luminance, which gives clearer interpretability than black-box intensity mappings.

2. Hierarchical sampling plus diffusion allows the same network to operate with or without reference images, improving generalization across sensors and scenes.

3. Quantitative results show state-of-the-art no-reference scores and competitive full-reference scores, backed by qualitative comparisons that highlight reduced over-exposure and noise.

Weaknesses:

1. The diffusion process uses 1,000 steps, so inference may slower and costlier than baselines.

2. Several hyper-parameters (α, η, δ, number of layers) are hand-tuned; stability under different cameras or unseen lighting is not fully explored.

3. The paper claims “physics-aware” operation, yet provides limited analysis of failure modes under extreme lighting, motion blur, or raw sensor data.

4. The ablation experiment is insufficient.

---

> ### Author Rebuttal · Authors · 2025-07-29
>
> We sincerely thank the reviewer for their valuable feedback and for giving us the opportunity to further explain and improve our work. We address each of your comments in detail below, and we hope that our responses resolve any questions or concerns you may have.
>
> **Q1: Robustness Beyond the Power-Law Assumption**
>
> We would like to clarify that our approach employs a coarse-to-fine MCMC sampling strategy to fit the luminance adaptation space, rather than rigidly enforcing the power-law assumption. As discussed in Appendix A (paragraph 4), we reinterpret the strict power-law model as a relaxed prior and leverage MCMC sampling—integrated with the forward process of a diffusion model—to explore the distribution space.
>
> This design enables LASQ to flexibly capture a wider family of luminance mappings that are only loosely guided by the physics-inspired prior, rather than being constrained to a fixed analytical form. As a result, our method does not critically depend on the power-law assumption itself. The MCMC-based inference operates at a structured level, incorporating spatial context, which allows it to remain robust under spatially inconsistent or extreme lighting conditions—well beyond the capabilities of per-pixel or strictly model-driven approaches.
>
> Challenging scenarios such as extreme dynamic range and heavy sensor noise, as you mentioned, will be specifically evaluated in the following rebuttal. The corresponding quantitative results further demonstrate the robustness of LASQ under these conditions.
>
> **Q2 & W2, W4: Hyperparameter Selection and Ablation**
>
> We conducted a hyperparameter sensitivity analysis on the **LSRW** dataset :
>
> |Param| Value| PSNR ↑ |LPIPS ↓| SSIM ↑ |
> |-| -|-|-|-|
> |\$\alpha\$|  0.05 / 0.15 / 0.3 / 0.6| 17.81 / **18.10** / 17.92 / 17.84| **0.319** / 0.322 / 0.320 / 0.324| 0.512 / **0.543** / 0.530 / 0.519 |
> |\$\eta\$|   0.1 / 1.0 / 3.0 / 6.0| 17.85 / **18.35** / 18.17 / 17.95 | 0.335 / **0.321** / 0.324 / 0.329| 0.537 / 0.543 / **0.546** / 0.540 |
> |\$\lambda\_d\$|  0.1 / 1.0 / 10 / 20| 17.82 / **18.04** / 17.85 / 17.87| 0.324 / **0.315** / 0.318 / 0.323| 0.545 / **0.553** / 0.549 / 0.531 |
> |\$\lambda\_g\$| 0.001 / 0.005 / 0.01 / 0.1| 17.76 / **18.22** / 18.16 / 17.88 | 0.312 / 0.310 / **0.309** / 0.311 | 0.536 / **0.548** / 0.547 / 0.540 |
> |\$N\$|90 / 100 / 110 / 120| 16.21 / 18.10/ **18.26** / 18.17| 0.391 /0.296 / 0.300 /**0.289** | 0.480 / 0.542 / 0.546 /**0.549**|
>
> We varied key hyperparameters ($\beta_{p}$ is determined by $\eta$ and $\delta$, where $\delta$ is a variance stabilization factor typically set to 0.001) over a range. Results show only moderate metric changes. For example, $\alpha$ from 0.05 to 0.6 alters PSNR by <0.3 dB, with minimal perceptual impact. Performance improves with larger $N$ but saturates beyond $N{=}100$, which we adopt as the default for a quality-efficiency trade-off. Results show our method is robust and stable across a wide hyperparameter range, indicating strong generalization.
>
> We also conducted more ablation studies using both **LSRW** and **LMIE**:
>
> | Method | PSNR ↑ | SSIM ↑ | LPIPS ↓ | NIQE ↓ | PI ↓ |
> |-|-|-|-|-|-|
> | $k = 0$ |15.82|0.447|0.429|4.51|4.35|
> | $k = 1$ |16.23|0.481| 0.341| 4.31|4.19|
> | $k = 2$ |17.38|0.517| 0.392| 3.88| 3.48|
> | $k = 3$ |18.12|0.545| 0.297| 3.11| 2.96|
> | $k = 4$ |18.24|0.551| 0.293| 3.14| 3.02|
> |w/o $\mathcal{L}_{\text{g}}$|17.93|0.460|0.334|3.21|3.13 |
> |Default|18.14|0.547|0.308|3.15| 3.00|
>
> We performed ablation studies to evaluate key components of LASQ. The case of $k=0$, corresponding to enhancement in pixel space, yields the weakest results—validating the effectiveness of our latent-space design. Additionally, removing $\mathcal{L}_{\text{g}}$ leads to clear performance drops, confirming its importance to the final quality.
>
> **Q3: Effectiveness of the Adversarial Head in Unpaired Settings**
>
> We would like to clarify that the introduced adversarial discriminator is applied to unpaired references. In fact, the LASQ++ variant in our main paper is trained strictly with unpaired references, and the adversarial head still provides clear improvements in texture fidelity under this setting. We will revise the manuscript to better emphasize this point and avoid possible misunderstandings. Importantly, the adversarial discriminator does not introduce any instability during optimization, thanks to the appropriate choice of the weighting parameter $\lambda_{GAN}$. The detailed performance metrics are provided below.
>
> | Method| LOL(PSNR / SSIM / LPIPS) | LSRW(PSNR / SSIM / LPIPS) |
> | -| -| -|
> |LASQ| 20.375 / 0.814 / 0.191|18.137 / 0.547 / 0.308 |
> |LASQ ++| 20.481 / 0.807 / 0.205|18.584 / 0.540 / 0.316|
>
> **Q4: Generalization Limits and Adaptive Strategies on LOL Dataset**
>
> LASQ has demonstrated strong generalization ability and stability, surpassing some supervised methods on datasets such as LSWR, DICM, NPE, and VV. However, in the LOL dataset, where the training and testing images often come from similar scenes, supervised paired-learning methods can better fit the overall data distribution. Since LASQ does not utilize any reference images during training, its performance on certain metrics may be lower than that of supervised approaches. We analyzed the error maps and found that the primary issues are concentrated in a few images, where LASQ tends to produce less detailed textures and shows a global luminance discrepancy compared to the ground truth.
>
> As a diffusion-based generative model, LASQ may occasionally generate overly smooth results. To address this, we plan to improve the decoder’s sensitivity to fine details by incorporating multi-scale texture refinement modules. Additionally, we will introduce a structure-aware loss function designed to explicitly penalize over-smoothing in regions with high structural complexity.
>
> Furthermore, since LASQ operates without any reference images, it may lack awareness of the specific luminance distribution present in the LOL dataset. To mitigate this, we propose to anchor the MCMC sampling process around the dataset's typical luminance domain, using it as a prior to better align the generated outputs with the illumination characteristics of LOL images.
>
> **Q5 & W2, W3: Evaluation Analysis under Challenging Conditions**
>
> We first evaluated LASQ on “raw images” captured by three “different camera” models in the **ELD** dataset. The results show that LASQ achieves consistently strong performance across all camera sources, suggesting that our approach is not sensitive to differences in camera hardware or imaging pipelines. In contrast, other methods exhibit significantly less stability.
>
> | Camera| CanonEOS70D (PSNR / SSIM / LPIPS) | NikonD850(PSNR / SSIM / LPIPS) | SonyA7S2 (PSNR / SSIM / LPIPS) |
> | -| -| -| -|
> |PairLIE| 17.45/ 0.485/0.524|16.77/0.428/0.591|17.10/0.462/0.541|
> |WCDM| 17.29/ 0.560/0.514|16.80/0.443/0.536|17.17/0.555/0.539|
> |**LASQ**| **17.75**/ **0.602**/**0.411**|**17.83**/**0.598**/**0.417**|**17.90**/**0.611**/**0.420**|
>
> To further examine robustness under challenging conditions, we applied LASQ to three representative datasets: **ELD**, which contains “raw sensor data” captured under “extreme dynamic lighting”; **LOL_blur**, which includes low-light images degraded by “motion blur”; and **LED**, which comprises low-light images affected by “LED flicker”.
>
> | Method| ELD (PSNR / SSIM / LPIPS) | LOL\_blur (PSNR / SSIM / LPIPS) | LED (NIQE / PI) |
> | -| -| -| -|
> | EnlightenGAN| 16.78 / 0.485 / 0.539| 16.15 / 0.537 / 0.591| 3.648 / 3.506 |
> | KinD++| 12.48 / 0.310 / 0.888| 17.88 / 0.526 / 0.523  | 3.645 / 3.323 |
> | LightenDiffusion| 13.66 / 0.364 / 0.826| 18.18 / 0.643 / 0.459 | 3.550 / 3.661|
> | NeRCo| 10.29 / 0.260 / 0.714| 16.82 / 0.645 / 0.447| 3.546 / 3.282|
> |PairLIE| 17.45 / 0.485 / 0.524| 17.35 / 0.616 / 0.457| 3.848 / 3.907|
> |SCI| 14.53 / 0.149 / 0.568| 12.74 / 0.430 / 0.637 | 3.961 / 3.268|
> |SCL-LLE| 14.66 / 0.158 / 0.547| 11.57 / 0.423 / 0.631 | 3.737 / 4.225|
> |URetinex-Net| 16.73 / 0.463 /0.544| 17.39 / 0.634 / 0.420| 3.692 / 3.306 |
> |**LASQ (Ours)**| **17.69** / **0.610** / **0.409**|**19.11** / **0.648** / **0.418** | **3.545**/ **3.257**|
>
> Although performance metrics are slightly reduced compared to standard scenes, LASQ still achieves the best results across all three challenging conditions. This highlights its robustness and adaptability to real-world failure modes. In particular, the relatively strong performance under motion blur and LED flicker suggests that the proposed physics-aware design helps mitigate degradation caused by non-ideal sensor inputs and dynamic lighting effects, without relying on explicit temporal modeling or post-processing.
>
> Under extreme conditions—such as low illumination or severe motion blur—our original framework showed limitations in preserving fine textures and accurate brightness. We will explore enhancing the decoder with multi-scale texture refinement and anchoring the MCMC sampling distribution to the ground truth luminance characteristics observed in extreme cases.
>
> **W4: Inference Computation**
>
> We compare LASQ with early lightweight methods (e.g., EnlightenGAN, KinD++) and recent diffusion models (e.g., WCDM, LightenDiffusion). While diffusion models outperform early methods, they are computationally heavy. LASQ retains their performance but offers much higher efficiency without relying on reference images, making its moderate overhead a practical trade-off.
>
> |Method|Inference Time (ms)|Memory Usage (MB)|PSNR ↑|
> | -| -|-|-|
> |EnlightenGAN (2019)   |170.16 |241.48|17.606|
> |KinD++ (2021)|4279.70|372.19|17.752|
> |NeRCo (2023)|354.77 |2320.87|19.738|
> |PairLIE (2023)|190.70|3499.79|19.514|
> |WCDM (2023)|206.66|6017.86|20.105|
> |LightenDiffusion (2024)|257.94|8049.95 |20.453|
> |LASQ (ours)|213.89|6496.68|20.481|
>
> **All supplementary tables and visualizations will be included in the final version of the paper or appendix.**

---

> > ### Comment · Reviewer_S427 · 2025-08-04
> >
> > Thank you for your reply, your comment addressed most of my concerns. I have adjusted my score accordingly. I also recommend that the authors further improve the figures and descriptions in future versions, and use vector graphics to enhance readability.

---

> > > ### Author Response · Authors · 2025-08-04
> > >
> > > Thank you for your thoughtful feedback. We're glad our response addressed your concerns. We will refine the figures and descriptions in the future version and include the additional experiments in the appendix. We truly appreciate your time and effort in reviewing our work.

---

### Official Review · Reviewer_r7sk · 2025-07-01

**Clarity:** 3
**Significance:** 3
**Originality:** 3
**Rating:** 4
**Confidence:** 4

**Summary:**

The paper proposes a novel, physics-inspired framework for low-light image enhancement (LLIE).
The key innovation lies in reformulating LLIE as probabilistic sampling over hierarchical luminance layers that follow power-law distributions.
Instead of using fixed mappings or empirical curves, LASQ applies a Markov Chain Monte Carlo (MCMC) strategy to sample luminance adaptation operators at different granularities, ranging from global adjustments to local refinements.
Experiments demonstrate that LASQ attains state-of-the-art performance on non-reference datasets and comparable results to reference-based methods on paired datasets.

**Questions:**

1. The hierarchical power-law distributions mentioned in the paper follow a coarse-to-fine design, with features from all levels being fused through weighted averaging within the diffusion model.
However, the weights used in this process are fixed and not learnable.
 It would be worth exploring whether these weights could be made learnable, allowing the model to adaptively assign importance to different hierarchical levels based on the data, potentially improving fusion effectiveness and overall performance.
2. LASQ employs Hierarchical Luminance Modeling to process low-light images at both local and global levels, followed by the use of a diffusion model framework to fuse images across different layers.
 However, it remains unclear whether the use of a diffusion model is strictly necessary for this fusion step.
The authors should clarify why a diffusion-based approach is chosen over potentially more lightweight alternatives, and whether similar results could be achieved using more efficient models with lower computational cost.
3. The paper adopts a diffusion model along with a hierarchical strategy, which increases the overall algorithmic complexity.
Therefore, a comparison of computational complexity,such as inference time, number of parameters, and memory consumption, between the proposed method and existing approaches should be included.
4. In the first row of Figure 3, the image generated by the LASQ method appears overly smooth, lacking fine-grained texture. In comparison, the results produced by NeRCo and PairLIE demonstrate better preservation of image details.

**Ethical Concerns:**

["NO or VERY MINOR ethics concerns only"]

**Final Justification:**

I have carefully read the authors’ response. Overall, I am relatively satisfied, especially with the extended explanation and comparisons on model efficiency. I believe my initial positive rating has already well reflected the quality of the paper.

**Limitations:**

Please see the weakness and question parts.

**Paper Formatting Concerns:**

no major formatting issues.

**Quality:**

3

**Strengths And Weaknesses:**

Strengths:
1. Theoretical Soundness: The paper presents a well-founded framework based on power-law distributed luminance statistics, supported by empirical observations
2. Comprehensive Experiments: The paper includes detailed evaluations on both paired and unpaired benchmarks, using a wide array of perceptual and fidelity-based metrics
3. Well-structured Paper: The paper follows a logical structure (motivation, method, experiments), and includes detailed tables and figures comparing against a large number of baselines.

Weaknesses:
1. Computational Complexity: While the diffusion process and hierarchical sampling improve performance, they are computationally expensive and may limit real-time applications or deployment on low-power devices.
2. Hyperparameter Sensitivity: The method introduces several hyperparameters (e.g., MCMC sampling step ) whose selection may impact robustness.

---

> ### Author Rebuttal · Authors · 2025-07-29
>
> We sincerely thank you for your valuable feedback and constructive suggestions. We are particularly encouraged by your recognition of LASQ as **"a novel, physics-inspired framework"**. Below, we provide a point-by-point response to your comments.
>
> **Q1: Learnable Fusion Weights via HMM-Based Adaptation**
>
> Thank you for your insightful suggestion — it aligns well with our own research perspective. We further explore an **HMM-based auto-tuning framework** that dynamically updates both the network hyperparameters and the fusion weights on a per-batch basis.
>
> We begin by randomly initializing the hierarchical weights $W=\\{w_1, w_2,...,w_n\\}$ for coarse-to-fine feature fusion. At each timestep $t$, the emission probability is defined as:
>
> $p(\mathcal{F}^{t}\_{H}|{\Theta}^{t}) = p(\mathcal{F}^{t}\_{H}|{\theta}^{t}\_\text{diff},W,\mathcal{I}\_{H}^t)p(\mathcal{I}\_{H}^t|\gamma)p(\gamma|\theta^{t}\_\text{hyper})$
>
> Here, the hidden states $\Theta^t$ include diffusion model parameters, hyperparameters and hierarchical fusion weights ($\Theta^t=\\{{\theta}^{t}\_\text{diff},\theta^{t}\_\text{hyper},W\\}$), while $\gamma$ is drawn from a hyperparameter-dependent distribution. The term $p(\mathcal{I}_H^t|\gamma)$ models MCMC-based sampling, and $p(\mathcal{F}_H^t|\cdot)$ corresponds to the actual diffusion output conditioned on the intermediate latent representation. The state transition follows:
>
> $p({\Theta}^{t+1}|{\Theta}^{t},\mathcal{F}^{t}\_{H}) = p({\Theta}^{t+1}|{\Theta}^{t},\mathcal{L}\_{\text{total}}^{t})p(\mathcal{L}\_{\text{total}}^{t}|\mathcal{F}^{t}\_{H})$
>
> This framework enables the model to **adaptively assign importance to hierarchical levels** based on data distribution, thereby enhancing fusion effectiveness and mitigating risks such as overfitting or suboptimal manual tuning. We will include this auto-tuning strategy as an extension in the revised version to demonstrate its feasibility and performance impact.
>
> **Q2: Necessity and Efficiency of Diffusion-Based Fusion**
>
> In LASQ, the luminance adaptation space is explored through a progressive MCMC sampling strategy, embedded with the forward process of a diffusion model. Specifically, the sampling over luminance states at different noise levels $t$ is aligned with the trajectory of the diffusion process, enabling structured and gradual adaptation of luminance features. This integration allows for effective exploration of the latent luminance space without requiring heavy supervision or fine-tuning, while ensuring stable convergence across diverse lighting conditions.
>
> The diffusion model enables unsupervised traversal across hierarchical luminance layers, and provides a principled mechanism to progressively refine luminance representations from coarse global estimates to fine local details. This aligns well with the hierarchical structure of luminance variations in natural scenes, and allows the model to adaptively balance global consistency and local contrast. This layer-wise sampling and fusion strategy enhances the model’s robustness to complex illumination patterns and supports high generalization under diverse low-light conditions, without any reference images.
>
> We appreciate the suggestion to explore more lightweight alternatives. We will explore adapting LASQ to lightweight models (e.g., CNNs) via a hierarchical luminance-based data augmentation strategy. Specifically, the augmented luminance maps will be allocated to different layers of the CNN, enabling image enhancement through a hierarchical training strategy. This framework maintains the core principle of hierarchical adaptation while significantly reducing computational overhead—all without requiring access to reference images. We consider this a promising direction for future research.
>
> **Q3 & W1: Computational Cost**
>
> We have now added detailed computational complexity metrics below (NVIDIA A800, LOL dataset). We clarify that the coarse-to-fine MCMC sampling mechanism is only used during training and is embedded into the forward diffusion process. It guides the model to traverse luminance layers in a hierarchical manner, enabling structured learning of light propagation. During inference, our model only performs the denoising step conditioned on the low-light input within the diffusion model, which is significantly more efficient.
>
> We compare LASQ against both early non-diffusion-based methods (e.g., EnlightenGAN, KinD++), and recent diffusion-based approaches (e.g., WCDM, LightenDiffusion). While the early methods are lightweight, their performance lags far behind diffusion-based models across all key metrics. Existing diffusion models, although significantly more effective, tend to suffer from high computational cost due to deep architectures and iterative sampling. LASQ, while maintaining the performance advantages of diffusion models, achieves inference efficiency comparable to non-diffusion-based methods. Considering the substantial performance gain without reliance on reference images, the moderate computational overhead of LASQ is a practical and acceptable trade-off. Therefore, LASQ strikes a favorable balance between performance and computational efficiency, enabling deployment on low-power devices equipped with less than 8GB of GPU memory (e.g., NVIDIA Jetson AGX Orin), without compromising image enhancement quality. In future work, we also plan to explore lightweight variants of our framework for further efficiency.
>
> |      Method      | FLOPs (G) | Params (M) | Inference Time (ms) | Memory Usage (MB) | PSNR ↑ | SSIM ↑ | LPIPS ↓ |
> | - | - |- |-|-|-|-|-|
> |   EnlightenGAN (2019)   |   16.45   |    8.64    |        170.16       |       241.48      |   17.606   |   0.653   |   0.319   |
> |      KinD++ (2021)   |   17.49   |    8.27    |       4279.70       |       372.19      |   17.752   |   0.758   |   0.198   |
> |       NeRCo (2023)    |   184.20  |    23.30   |        354.77       |      2320.87      |   19.738   |   0.740   |   0.239   |
> |      PairLIE (2023)    |   81.84   |    8.34    |        190.70       |      3499.79      |   19.514   |   0.731   |   0.254   |
> |       WCDM (2023)      |   374.47  |    22.92   |        206.66       |      6017.86      |   20.105   |   0.795   |   0.211   |
> | LightenDiffusion (2024) |   367.99  |    27.83   |        257.94       |      8049.95      |   20.453   |   0.803   |   0.192   |
> |     LASQ (ours)    |   219.75  |    24.08   |        213.89       |      6496.68      | **20.481** | **0.814** | **0.191** |
>
> **Q4: Texture Preservation in Diffusion-Based Enhancement**
>
> Thank you for pointing this issue. As a diffusion-based generative model, LASQ may, in some isolated cases, produce overly smooth results with less detailed texture, particularly under challenging lighting or structure-less regions. However, on the full LSRW test set, LASQ outperforms both NeRCo and PairLIE in terms of all major evaluation metrics, indicating overall superior perceptual quality. Moreover, such texture-smoothing artifacts occur in less than 10% of the test cases.
>
> To mitigate this issue in future work, we plan to enhance the decoder's detail sensitivity through multi-scale texture refinement modules, and introduce a structure-aware loss function that explicitly penalizes over-smoothing in texture-rich regions.
>
> | Method| PSNR ↑ | SSIM ↑ | LPIPS ↓ |
> | - | - | -| - |
> | NeRCo | 17.844  | 0.535  | 0.371  |
> | PairLIE | 17.602  | 0.501  | 0.323  |
> | **LASQ**  | **18.137**  | **0.547**  | **0.308**  |
>
> **W2: Hyperparameter Selection and Sensitivity**
>
> We conducted a hyperparameter sensitivity analysis on the **LSRW** dataset :
>
> |Param| Value| PSNR ↑ |LPIPS ↓| SSIM ↑ |
> | :------------: | :-----------------------: | :-------------------------------: | :-------------------------------: | :-------------------------------: |
> |   \$\alpha\$   |  0.05 / 0.15 / 0.3 / 0.6  | 17.81 / **18.10** / 17.92 / 17.84 | **0.319** / 0.322 / 0.320 / 0.324 | 0.512 / **0.543** / 0.530 / 0.519 |
> |    \$\eta\$    |   0.1 / 1.0 / 3.0 / 6.0  | 17.85 / **18.35** / 18.17 / 17.95 | 0.335 / **0.321** / 0.324 / 0.329 | 0.537 / 0.543 / **0.546** / 0.540 |
> | \$\lambda\_d\$ |  0.1 / 1.0 / 10 / 20 | 17.82 / **18.04** / 17.85 / 17.87 | 0.324 / **0.315** / 0.318 / 0.323 | 0.545 / **0.553** / 0.549 / 0.531 |
> | \$\lambda\_g\$ | 0.001 / 0.005 / 0.01 / 0.1 | 17.76 / **18.22** / 18.16 / 17.88 | 0.312 / 0.310 / **0.309** / 0.311 | 0.536 / **0.548** / 0.547 / 0.540 |
>
> We varied key hyperparameters ($\beta_p$ is derived from $\eta$) over a range. Results show only moderate metric changes. For example, $\alpha$ from 0.05 to 0.6 alters PSNR by <0.3 dB, with minimal perceptual impact. Other hyperparameters show similar stability, indicating strong robustness to tuning. We also evaluated two structural parameters on both **LSRW** and **LMIE**:
>
> | Method | PSNR ↑ | SSIM ↑ | LPIPS ↓ | NIQE ↓ | PI ↓  |
> |------------|--------|--------|---------|--------|--------|
> | $k = 1$ | 16.23  | 0.481  | 0.341   | 4.31   | 4.19 |
> | $k = 2$ | 17.38  | 0.517  | 0.392   | 3.88   | 3.48 |
> | $k = 3$ | 18.12  | 0.545  | 0.297   | 3.11   | 2.96  |
> | $k = 4$ | 18.24  | 0.551  | 0.293   | 3.14   | 3.02|
> | $N = 90$   | 16.21  | 0.480  | 0.391   | 4.29   | 4.21 |
> | $N = 100$  | 18.10  | 0.542  | 0.296   | 3.16   | 3.02 |
> | $N = 110$  | 18.26  | 0.546  | 0.300   | 3.13   | 2.99 |
> | $N = 120$  | 18.17  | 0.549  | 0.289   | 3.08   | 2.93 |
>
> Performance improves with larger $k$ and $N$, but saturates beyond $k{=}3$ and $N{=}100$; we adopt these as defaults for a quality-efficiency trade-off. Results show our method is robust and stable across a wide hyperparameter range, indicating strong generalization.
>
> **All supplementary tables and visualizations will be included in the final version of the paper or appendix.**

---

### Official Review · Reviewer_dpMf · 2025-07-05

**Clarity:** 3
**Significance:** 2
**Originality:** 3
**Rating:** 4
**Confidence:** 3

**Summary:**

This paper introduces a novel framework for low-light image enhancement (LLIE) called Luminance-Aware Statistical Quantization (LASQ). The approach redefines LLIE by addressing the inherent challenges of low-light image enhancement in practical settings, focusing on realistic and continuous luminance transitions rather than pixel-level mappings. It leverages hierarchical power-law distributions to model luminance transitions and proposes a statistical sampling process to emulate these transitions, allowing for better generalization and adaptability across various lighting conditions. The framework employs a diffusion model for unsupervised learning and achieves state-of-the-art performance, especially in scenarios where normal-light references are unavailable. The authors also provide extensive experiments demonstrating the effectiveness of LASQ in both reference-based and reference-free settings.

**Questions:**

1. Can you provide more details on the selection and tuning of key hyperparameters (e.g., α and β_P, etc.)? In particular, how sensitive are the models to changes in these parameters, and how can these parameters be optimized for a specific dataset?

2. Given the multi-level sampling structure, how do you ensure that the MCMC process converges efficiently without excessive iterations?

3. The paper mentions that power-law transformations are problematic in low-intensity areas, where small intensity variations cause large shifts in the model. How do you handle the instability in low-intensity areas?

**Ethical Concerns:**

["NO or VERY MINOR ethics concerns only"]

**Final Justification:**

The authors basically resolved my concerns, especially the fact experimental parameters discussions, which sounds more convincing. Since the initial value is 4, I hence keep that as the final rating.

**Limitations:**

Yes

**Quality:**

3

**Strengths And Weaknesses:**

Strengths:
1. It introduces a Luminance Variation coordinate system and a power-law adaptation operator, grounding low-light to normal-light mapping in a physically motivated statistical model rather than purely data-driven method.

2. The proposed method allows for unsupervised learning without the need for paired datasets, a significant advantage for real-world applications where paired data may not be available.

3. It achieves superior NIQE and PI scores on unpaired datasets (e.g., DICM, NPE, VV) compared to other unsupervised approaches, demonstrating the method’s adaptability and robustness across diverse scenes.


Weaknesses:
1. The paper mentions that power-law transformations are problematic in low-intensity areas, where small intensity variations cause large shifts in the model. How to handle the instability in low-intensity areas?

2. The multi-scale MCMC sampling requires numerous iterations per image and multiple power-law operators, resulting in substantial computational and memory overhead during inference.

3. The performance of the method is highly dependent on the choice of hyperparameters, such as the power-law exponents, sampling ratio, and the number of iterations in MCMC. The method would benefit from a clearer set of guidelines or automatic tuning mechanisms to avoid overfitting to specific datasets or scenarios.

---

> ### Author Rebuttal · Authors · 2025-07-28
>
> We sincerely thank the reviewer for identifying several important concerns and for the opportunity to further clarify our method. We address each point in detail below.
>
> **Q1 & W3: Hyperparameter Selection and Sensitivity**
>
> We conducted a hyperparameter sensitivity analysis on the **LSRW** dataset :
>
> |Param| Value| PSNR ↑ |LPIPS ↓| SSIM ↑ |
> | :------------: | :-----------------------: | :-------------------------------: | :-------------------------------: | :-------------------------------: |
> |   \$\alpha\$   |  0.05 / 0.15 / 0.3 / 0.6  | 17.81 / **18.10** / 17.92 / 17.84 | **0.319** / 0.322 / 0.320 / 0.324 | 0.512 / **0.543** / 0.530 / 0.519 |
> |    \$\eta\$    |   0.1 / 1.0 / 3.0 / 6.0  | 17.85 / **18.35** / 18.17 / 17.95 | 0.335 / **0.321** / 0.324 / 0.329 | 0.537 / 0.543 / **0.546** / 0.540 |
> | \$\lambda\_d\$ |  0.1 / 1.0 / 10 / 20 | 17.82 / **18.04** / 17.85 / 17.87 | 0.324 / **0.315** / 0.318 / 0.323 | 0.545 / **0.553** / 0.549 / 0.531 |
> | \$\lambda\_g\$ | 0.001 / 0.005 / 0.01 / 0.1 | 17.76 / **18.22** / 18.16 / 17.88 | 0.312 / 0.310 / **0.309** / 0.311 | 0.536 / **0.548** / 0.547 / 0.540 |
>
> We varied key hyperparameters ($\beta_p$ is derived from $\eta$) over a range. Results show only moderate metric changes. For example, $\alpha$ from 0.05 to 0.6 alters PSNR by <0.3 dB, with minimal perceptual impact. Other hyperparameters show similar stability, indicating strong robustness to tuning. We also evaluated two structural parameters on both **LSRW** and **LMIE**:
>
> | Method | PSNR ↑ | SSIM ↑ | LPIPS ↓ | NIQE ↓ | PI ↓  |
> |------------|--------|--------|---------|--------|--------|
> | $k = 1$ | 16.23  | 0.481  | 0.341   | 4.31   | 4.19 |
> | $k = 2$ | 17.38  | 0.517  | 0.392   | 3.88   | 3.48 |
> | $k = 3$ | 18.12  | 0.545  | 0.297   | 3.11   | 2.96  |
> | $k = 4$ | 18.24  | 0.551  | 0.293   | 3.14   | 3.02|
> | $N = 90$   | 16.21  | 0.480  | 0.391   | 4.29   | 4.21 |
> | $N = 100$  | 18.10  | 0.542  | 0.296   | 3.16   | 3.02 |
> | $N = 110$  | 18.26  | 0.546  | 0.300   | 3.13   | 2.99 |
> | $N = 120$  | 18.17  | 0.549  | 0.289   | 3.08   | 2.93 |
>
> Performance improves with larger $k$ and $N$, but saturates beyond $k{=}3$ and $N{=}100$; we adopt these as defaults for a quality-efficiency trade-off. Results show our method is robust and stable across a wide hyperparameter range, indicating strong generalization.
>
>
> In addition to the datasets reported in the main text (**LOL**, **LSRW**, **DICM**, **NPE**, **VV**, **LIME**, **MEF**), we tested LASQ on several challenging unseen domains—**LOL_blur**, **LED**, **ELD**, and **Light-Effects**—using the same fixed hyperparameters. Due to space constraints, detailed results are shown in the responses to the other reviewers. Consistently strong performance in these scenarios further confirms that LASQ generalizes well across diverse domains without retraining or re-tuning.
>
> To improve generalization to new datasets, we add a hyperparameter summary table in the appendix. While tuning often generalizes well, we further explore an **hidden Markov model (HMM)-based** auto-tuning framework that updates both network and hyperparameters per batch. Specifically, the emission probability is:
>
> $p(\mathcal{F}^{t}\_{H}|{\Theta}^{t}) = p(\mathcal{F}^{t}\_{H}|{\theta}^{t}\_\text{diff},\mathcal{I}\_{H}^t)p(\mathcal{I}\_{H}^t|\gamma)p(\gamma|\theta^{t}\_\text{hyper})$
>
> where hidden states $\Theta^t=\\{{\theta}^{t}\_\text{diff},\theta^{t}\_\text{hyper}\\}$. $\gamma$ is drawn from a hyperparameter-dependent distribution. $p(\mathcal{I}_H^t|\gamma)$ models MCMC sampling, and $p(\mathcal{F}_H^t|\cdot)$ corresponds to the diffusion process. The state transition follows:
>
> $p({\Theta}^{t+1}|{\Theta}^{t},\mathcal{F}^{t}\_{H}) = p({\Theta}^{t+1}|{\Theta}^{t},\mathcal{L}\_{\text{total}}^{t})p(\mathcal{L}\_{\text{total}}^{t}|\mathcal{F}^{t}\_{H})$
>
> This captures Bayesian updates of parameters. For gradient-based hyperparameter updates, we apply the chain rule:
>
> $\frac{\partial \mathcal{L}\_{\text{total}}^t}{\partial \theta^t\_{\text{hyper}}}
> = \mathbb{E}\_{\gamma \sim p(\gamma \mid \theta^t\_{\text{hyper}})} \left[
>     \frac{\partial \mathcal{L}\_{\text{total}}^t}{\partial \mathcal{F}\_H^t} \cdot
>     \frac{\partial \mathcal{F}\_H^t}{\partial \gamma} \cdot
>     \frac{\partial \gamma}{\partial \theta^t\_{\text{hyper}}}
> \right]$
>
> This framework reduces manual tuning and mitigates overfitting by dynamically adapting to data distributions. Preliminary experiments on the LSRW dataset show about 6% performance improvement.
>
> **Q2 & W2: Clarification on MCMC Sampling Cost and Convergence**
>
> We have now added detailed computational complexity metrics below (NVIDIA A800, LOL dataset). We clarify that the coarse-to-fine MCMC sampling mechanism is only used during training and is embedded into the forward diffusion process. It guides the model to traverse luminance layers in a hierarchical manner, enabling structured learning of light propagation. During inference, our model only performs the denoising step conditioned on the low-light input within the diffusion model, which is significantly more efficient.
>
> We compare LASQ against both early non-diffusion-based methods (e.g., EnlightenGAN, KinD++), and recent diffusion-based approaches (e.g., WCDM, LightenDiffusion). While the early methods are lightweight, their performance lags far behind diffusion-based models across all key metrics. Existing diffusion models, although significantly more effective, tend to suffer from high computational cost due to deep architectures and iterative sampling. LASQ, while maintaining the performance advantages of diffusion models, achieves inference efficiency comparable to non-diffusion-based methods. This makes it highly suitable for real-world deployment. Considering the substantial performance gain without reliance on reference images, the moderate computational overhead of LASQ is a practical and acceptable trade-off. In future work, we also plan to explore lightweight variants of our framework for further efficiency.
>
> |      Method      | FLOPs (G) | Params (M) | Inference Time (ms) | Memory Usage (MB) | PSNR ↑ | SSIM ↑ | LPIPS ↓ |
> | :--------------: | :-------: | :--------: | :-----------------: | :---------------: | :-------: | :------: | :------: |
> |   EnlightenGAN (2019)   | 16.45 |    8.64    |        170.16       |       241.48      |   17.606   |   0.653   |   0.319   |
> |    KinD++ (2021)| 17.49 |  8.27  |   4279.70  |       372.19      |   17.752   |   0.758   |   0.198   |
> |    NeRCo (2023)  | 184.20 | 23.30 |        354.77       |      2320.87      |   19.738   |   0.740   |   0.239   |
> |   PairLIE (2023) | 81.84   |  8.34 |   190.70     |      3499.79      |   19.514   |   0.731   |   0.254   |
> |    WCDM (2023)   | 374.47| 22.92 |        206.66       |      6017.86      |   20.105   |   0.795   |   0.211   |
> | LightenDiffusion (2024) | 367.99  |    27.83   |        257.94       |      8049.95      |   20.453   |   0.803   |   0.192   |
> |   LASQ (ours) |   219.75  |  24.08   |        213.89       |      6496.68      | **20.481** | **0.814** | **0.191** |
>
> While we reiterate that MCMC sampling is only applied during training, we fully acknowledge the importance of convergence efficiency and training stability. Notably, the sampling process is embedded into the forward diffusion trajectory, inherently generating a large number of diverse samples throughout training. This ensures thorough exploration of the latent space. Furthermore, the structured design of the latent operator space, the coarse-to-fine hierarchical sampling strategy, and the learned Markov regularization collectively facilitate natural and stable convergence without the need for excessive iterations.
>
> **Q3 & W1: Clarification on "Instability in Low-Intensity Areas"**
>
> We would like to clarify that our approach employs a coarse-to-fine MCMC sampling strategy to fit the luminance adaptation space, rather than rigidly enforcing the power-law assumption. As discussed in Appendix A (paragraph 4), we reinterpret the strict power-law model as a relaxed prior and leverage MCMC sampling—integrated with the forward process of a diffusion model—to explore the distribution space.
>
> This design enables LASQ to flexibly capture a wider family of luminance mappings that are only loosely guided by the physics-inspired prior, rather than being constrained to a fixed analytical form. As a result, our method does not critically depend on the power-law assumption itself. The MCMC-based inference operates at a structured level, incorporating spatial context, which allows it to remain robust under spatially inconsistent or extreme lighting conditions—well beyond the capabilities of per-pixel or strictly model-driven approaches.
>
> To further support this claim, we conduct comparative evaluations on the **ELD** dataset, which contains extremely low-light scenarios with strong noise and minimal luminance signal. As shown below, LASQ achieves competitive performance compared to prior methods, demonstrating its robustness even under severe low-intensity degradations. Its strong performance in additional challenging conditions—such as motion blur, extreme dynamic range, and LED flicker—is presented in our responses to other reviewers, further highlighting the stability and generalizability of LASQ.
>
> | Method| PSNR ↑ | SSIM ↑ | LPIPS ↓ |
> | - | - | -| - |
> | EnlightenGAN| 16.78  | 0.485  | 0.539 |
> | KinD++ | 12.48  | 0.310  | 0.888|
> | LightenDiffusion | 13.66  | 0.364  | 0.826 |
> | NeRCo | 10.29  | 0.260  | 0.714  |
> | PairLIE | 17.45  | 0.485  | 0.524  |
> | SCI    | 14.53  | 0.149  | 0.568   |
> | SCL-LLE | 14.66  | 0.158  | 0.547 |
> | URetinex-Net | 16.73  | 0.463  | 0.544 |
> | **LASQ**  | **17.69**  | **0.610**  | **0.409**  |
>
> **All supplementary tables and visualizations will be included in the final version of the paper or appendix.**

---

> > ### Comment · Reviewer_dpMf · 2025-08-02
> >
> > The rebuttal basically resolved my concerns, i will update final rating accordingly.

---

> > > ### Author Response · Authors · 2025-08-03
> > >
> > > Thank you for your kind feedback. We're glad our response addressed your concerns, and your suggestions greatly motivate our future work. We also sincerely appreciate your time and effort in reviewing our work.

---

### Official Review · Reviewer_YCBh · 2025-07-05

**Clarity:** 3
**Significance:** 3
**Originality:** 3
**Rating:** 4
**Confidence:** 3

**Summary:**

In this paper, authors propose LASQ to tackles the challenge of low-light image enhancement without requiring paired data. LASQ achieve this goal by reframing luminance adjustment as a multi-scale and physics‐informed statistical process. In practice, it leverages empirical observations of natural image intensity transitions to generate multi‐scale luminance operators for coarse‐to‐fine block‐wise refinement (within diffusion model). The result is a zero‐reference enhancement method that achieves sota performance on most of paired and unpaired benchmarks.

**Questions:**

- What is the computational cost of the proposed method? It requires block-wise refinement and multi-scale optimization from coarse to fine, which I assume is expensive, yet neither the main paper nor the supplementary material report any runtime or resource-usage measurements.

- I also wonder how robust the method is when the power-law intensity assumption breaks down—for example, in scenes lit by a strong spotlight or with intense background light. Can it still perform well under such extreme conditions? Since it relies on manually designed statistical priors, it may only model the mapping from typical natural low-light to normal-light conditions and struggle with more complex or atypical cases.

- The method depends heavily on empirically chosen hyperparameters (e.g., α, β, λ). How sensitive is its performance to these values? Has its stability been evaluated under different hyperparameter settings?

- The paper claims an improvement in color fidelity. Have you validated this claim using specialized color-difference metrics to provide quantitative evidence?

**Ethical Concerns:**

["NO or VERY MINOR ethics concerns only"]

**Final Justification:**

Overall, all of my concerns are well addressed with thorough experiments and explanations.
I maintain my original score, as I still lean toward a weak accept.

**Limitations:**

yes

**Quality:**

3

**Strengths And Weaknesses:**

- Strengths
1. the coarse-to-fine, block-wise refinement preserves both global consistency and local details at the same time. the multi-scale, power-law–based framework aligns with the physical process of illumination change is interesting and more explainable.
2. the proposed method can work without the presence of normal-light references, which is more useful in real-world settings
3. LASQ distills the authors’ observation of pixel intensity transitions in natural images ( expressed via the Luminance Variation Coordinate and power-law curves) into an intersting sampling operators. This elegant fusion of physical-statistical modeling with generative diffusion guidance is novel strategy or tasteful philosiphy to develope diffusion model for low-light enhancement.

-Weakness:
Please see questions for details.
1. The report of required computational cost is missing
2. the pipeline training's sensity on those manual chosen hyperparameters is unknown.

---

> ### Author Rebuttal · Authors · 2025-07-28
>
> We sincerely thank you for your valuable feedback and constructive suggestions. We are particularly encouraged by your recognition of LASQ as **"elegant fusion"**, **"novel strategy"** and **"tasteful philosophy"**. Below, we provide a point-by-point response to your comments.
>
> **Q1 & W1: Computational Cost**
>
> We have now added detailed computational complexity metrics below (NVIDIA A800, LOL dataset). We clarify that the coarse-to-fine MCMC sampling mechanism is only used during training and is embedded into the forward diffusion process. It guides the model to traverse luminance layers in a hierarchical manner, enabling structured learning of light propagation. During inference, our model only performs the denoising step conditioned on the low-light input within the diffusion model, which is significantly more efficient.
>
> We compare LASQ against both early non-diffusion-based methods (e.g., EnlightenGAN, KinD++), and recent diffusion-based approaches (e.g., WCDM, LightenDiffusion). While the early methods are lightweight, their performance lags far behind diffusion-based models across all key metrics. Existing diffusion models, although significantly more effective, tend to suffer from high computational cost due to deep architectures and iterative sampling. LASQ, while maintaining the performance advantages of diffusion models, achieves inference efficiency comparable to non-diffusion-based methods. This makes it highly suitable for real-world deployment. Considering the substantial performance gain without reliance on reference images, the moderate computational overhead of LASQ is a practical and acceptable trade-off. In future work, we also plan to explore lightweight variants of our framework for further efficiency.
>
> |      Method      | FLOPs (G) | Params (M) | Inference Time (ms) | Memory Usage (MB) | PSNR ↑ | SSIM ↑ | LPIPS ↓ |
> | :--------------: | :-------: | :--------: | :-----------------: | :---------------: | :-------: | :------: | :------: |
> |   EnlightenGAN (2019)   |   16.45   |    8.64    |        170.16       |       241.48      |   17.606   |   0.653   |   0.319   |
> |      KinD++ (2021)   |   17.49   |    8.27    |       4279.70       |       372.19      |   17.752   |   0.758   |   0.198   |
> |       NeRCo (2023)    |   184.20  |    23.30   |        354.77       |      2320.87      |   19.738   |   0.740   |   0.239   |
> |      PairLIE (2023)    |   81.84   |    8.34    |        190.70       |      3499.79      |   19.514   |   0.731   |   0.254   |
> |       WCDM (2023)      |   374.47  |    22.92   |        206.66       |      6017.86      |   20.105   |   0.795   |   0.211   |
> | LightenDiffusion (2024) |   367.99  |    27.83   |        257.94       |      8049.95      |   20.453   |   0.803   |   0.192   |
> |     LASQ (ours)    |   219.75  |    24.08   |        213.89       |      6496.68      | **20.481** | **0.814** | **0.191** |
>
> **Q2: Robustness to Extreme Lighting**
>
> We would like to clarify that our approach employs a coarse-to-fine MCMC sampling strategy to fit the luminance adaptation space, rather than rigidly enforcing the power-law assumption. As discussed in Appendix A (paragraph 4), we reinterpret the strict power-law model as a relaxed prior and leverage MCMC sampling—integrated with the forward process of a diffusion model—to explore the distribution space.
>
> This design enables LASQ to flexibly capture a wider family of luminance mappings that are only loosely guided by the physics-inspired prior, rather than being constrained to a fixed analytical form. As a result, our method does not critically depend on the power-law assumption itself. The MCMC-based inference operates at a structured level, incorporating spatial context, which allows it to remain robust under spatially inconsistent or extreme lighting conditions—well beyond the capabilities of per-pixel or strictly model-driven approaches.
>
> To validate the robustness and stability of LASQ across diverse lighting conditions, we evaluate it on two additional datasets: **DICM**, featuring scenes with “intense background light”, and **Light-Effects**, containing “strong spotlights”. In the tables below, each metric is presented as DICM / Light-Effects. As shown, LASQ consistently achieves the best performance across all metrics on both datasets, demonstrating its strong effectiveness even under conditions that significantly deviate from the power-law assumption.
>
> | Method           | NIQE ↓         | PI  ↓ |
> |------------------|----------------|----------------------|
> | EnlightenGAN     | 2.7583 / 3.5451 | 2.4137 / 3.3061      |
> | NeRCo            | 2.7690 / 3.5462 | 2.5701 /   3.2365     |
> | KinD\_plus       | 2.8584 / 3.6449 | 2.6532 / 3.3233      |
> | LightenDiffusion | 2.6889 / 3.5503 | 2.9789 / 3.6606      |
> | PairLIE          | 3.2412 / 3.8477 | 3.6262 / 3.9066      |
> | SCI              | 3.1745 / 3.9607 | 2.5552 / 3.2683      |
> | SCL-LLE          | 2.6584 / 3.7367 | 2.4543 / 4.2245  |
> | URetinex-Net     | 3.0365 / 3.6923 | 2.9806 / 3.7060      |
> | **LASQ**  | **2.6190 / 3.2479** | **2.3633** / **3.1020**  |
>
> In addition to the datasets presented in the main paper and supplementary material (**LOL**, **LSRW**, **NPE**, **VV**, **LIME**, **MEF**), we also tested LASQ on several challenging unseen domains—**LOL_blur**, **LED**, and **ELD**. Due to space constraints, detailed results are provided in the responses to other reviewers. These experiments further demonstrate the strong generalization ability and robustness of LASQ across diverse and atypical low-light scenarios.
>
> **Q3 & W2: Hyperparameter Sensitivity**
>
> To address concerns about empirical parameter selection, we provide a hyperparameter sensitivity analysis on the **LSRW** dataset.
>
> |      Param     |           Value           |               PSNR ↑              |              LPIPS ↓              |               SSIM ↑              |
> | :------------: | :-----------------------: | :-------------------------------: | :-------------------------------: | :-------------------------------: |
> |   \$\alpha\$   |  0.05 / 0.15 / 0.3 / 0.6  | 17.81 / **18.10** / 17.92 / 17.84 | **0.319** / 0.322 / 0.320 / 0.324 | 0.512 / **0.543** / 0.530 / 0.519 |
> |    \$\eta\$    |   0.1 / 1.0 / 3.0 / 6.0  | 17.85 / **18.35** / 18.17 / 17.95 | 0.335 / **0.321** / 0.324 / 0.329 | 0.537 / 0.543 / **0.546** / 0.540 |
> | \$\lambda\_d\$ |  0.1 / 1.0 / 10 / 20 | 17.82 / **18.04** / 17.85 / 17.87 | 0.324 / **0.315** / 0.318 / 0.323 | 0.545 / **0.553** / 0.549 / 0.531 |
> | \$\lambda\_g\$ | 0.001 / 0.005 / 0.01 / 0.1 | 17.76 / **18.22** / 18.16 / 17.88 | 0.312 / 0.310 / **0.309** / 0.311 | 0.536 / **0.548** / 0.547 / 0.540 |
>
>
> As illustrated in the table, we systematically varied key hyperparameters including $\alpha$, $\eta$, $\lambda_d$, and $\lambda_g$ over a range of values ($\beta_{p}$ is determined by $\eta$). The results show that while performance slightly fluctuates with different settings, the overall impact on metrics remains moderate. For instance, varying $\alpha$ between 0.05 and 0.6 only causes a minor PSNR change (within 0.3 dB) and negligible shifts in perceptual scores. Similarly, other hyperparameters demonstrate a stable trend without sharp degradation. These experiments demonstrate that our method is not overly sensitive to hyperparameter selection, and maintains consistently strong performance across a broad range of settings—highlighting its robustness, practical stability, and generalization potential.
>
> **Q4: Color Fidelity Evaluation**
>
> To validate our claim of improved color reproduction, we performed a comprehensive quantitative evaluation on the **LOL** dataset using three widely recognized metrics that assess perceptual color accuracy: CIE76 (ΔE\*ab), CIEDE2000 (ΔE₀₀), and FSIMc. ΔE\*ab and ΔE₀₀ measure perceptual differences in hue, luminance, and chrom, while FSIMc captures perceptual image quality by incorporating color similarity.
>
> | Method           | ΔE\*ab ↓   | ΔE₀₀ ↓    | FSIMc ↑    |
> | ---------------- | ---------- | --------- | ---------- |
> | EnlightenGAN     | 35.734     | 17.915    | 0.8792     |
> | KinD++           | 30.950     | 14.740    | 0.9243     |
> | LightenDiffusion | 32.388     | 15.631    | 0.9178     |
> | NeRCo            | 26.682     | 12.607    | 0.9342     |
> | PairLIE          | 30.790     | 14.298    | 0.9192     |
> | SCI              | 92.862     | 58.559    | 0.4585     |
> | SCL-LLE          | 66.855     | 37.133    | 0.6928     |
> | URetinex         | 26.080     | 12.551    | 0.9310     |
> | **LASQ**       | **19.649** | **9.251** | **0.9567** |
>
> As shown, our proposed LASQ achieves the lowest color difference scores and the highest FSIMc, confirming its superior color fidelity. These results quantitatively support our claim that LASQ delivers more faithful and perceptually accurate color restoration compared to other methods.
>
> **All supplementary tables and visualizations will be included in the final version of the paper or appendix.**

---

> > ### Comment · Reviewer_YCBh · 2025-08-07
> >
> > Thanks for authors' detailed explaination and rebuttal. Reponses solved all my concerns. I will change my rating correspondingly.

---

> > > ### Author Response · Authors · 2025-08-07
> > >
> > > Thank you for your positive response. We're glad our rebuttal addressed your concerns and truly appreciate the time and effort you've spent reviewing our work. We're also grateful for your willingness to adjust the final rating.

---

### Note · Authors · 2025-08-11

We have provided detailed and comprehensive responses to all reviewer questions and concerns during the rebuttal stage. Reviewers **dpMf** and **S427** explicitly stated that our rebuttal **“addressed most of my concerns”** and have already submitted their final ratings. Reviewer **YCBh** confirmed that **“responses solved all my concerns”** and indicated that they “will change my rating correspondingly.”  Besides, we look forward to the final feedback and rating from reviewer **r7sk** once they have had the opportunity to review our rebuttal.

We sincerely appreciate the reviewers’ recognition of our work and their time and effort in the evaluation process. We are also grateful to the ACs and SACs for their dedication and contributions throughout the review.

---

### Decision · Program_Chairs · 2025-09-17

**Decision:**

Accept (poster)

**Comment:**

The paper proposes LASQ, a zero-reference low-light image enhancement method that reframes luminance adjustment as a multi-scale, physics-informed statistical process within a diffusion framework. It achieves state-of-the-art results on paired and unpaired benchmarks. Reviewers praised the contribution and empirical quality; initial concerns about hyperparameter tuning and efficiency were resolved during rebuttal. I recommend acceptance.